# Group invariant machine learning by fundamental domain projections

**Benjamin Aslan**[*]                                   UCAHBAS@UCL.AC.UK
**University College London, 25 Gordon Street, WC1H 0AY, UK**

**Daniel Platt**[*]                                   DANIEL.1.PLATT@KCL.AC.UK
**King's College London, Strand Campus, WC2R 2LS, UK**

**David Sheard**[*]                                   DAVID.SHEARD.17@UCL.AC.UK
**University College London, 25 Gordon Street, WC1H 0AY, UK**

**Editor:** Sophia Sanborn, Christian Shewmake, Simone Azeglio, Arianna Di Bernardo, Nina Miolane

## Abstract

We approach the well-studied problem of supervised group invariant and equivariant machine learning from the point of view of geometric topology. We propose a novel approach using a pre-processing step, which involves projecting the input data into a geometric space which parametrises the orbits of the symmetry group. This new data can then be the input for an arbitrary machine learning model (neural network, random forest, support-vector machine etc). We give an algorithm to compute the geometric projection, which is efficient to implement, and we illustrate our approach on some example machine learning problems (including the well-studied problem of predicting Hodge numbers of CICY matrices), finding an improvement in accuracy versus others in the literature.

**Keywords:** Group invariant, group equivariant, geometric deep learning, fundamental domain, geometric topology

## 1. Introduction

Many tasks in machine learning can be understood as approximating a function $\alpha\colon X \to Y$ between a feature space and an output space. Typically, these may be subsets of $\mathbb{R}^n$, but could be more complicated like Riemannian manifolds. We consider the problem in the presence of *symmetries*—more precisely, suppose a group $G$ that acts on $X$ on the left, and $\alpha$ satisfies the *invariance property*

$$\alpha(g \cdot x) = \alpha(x) \text{ for all } x \in X, g \in G. \tag{1}$$

A simple example is recognising a single handwritten digit which may have been rotated by $90°$, $180°$, or $270°$, so the problem is invariant under the action of $\mathbb{Z}_4$.

Machine learning models such as neural networks or random forests can approximate $\alpha$ but the resulting function $\beta$ will not generally be $G$-invariant. The key task is to define machine learning algorithms producing functions $\beta\colon X \to Y$ which are necessarily invariant.

---

[*] equal contribution

## 1.1. Previous work

Machine learning models which are invariant (or equivariant) under the action of a group $G$ have been extensively studied in the literature. In (40), Yarotsky distinguishes two different approaches to the problem: *symmetrisation based* and *intrinsic* approaches. The first involves averaging some non $G$-invariant model over the action of $G$ to produce an (approximately) $G$-invariant model; whereas intrinsic approaches involve designing the model to be $G$-invariant *a priori* by imposing conditions coming from the group action.

A standard symmetrisation based approach is *data augmentation*, which was used in early works such as (25), and is surveyed in (7). It involves increasing the size of the training data $D_{\text{train}} = \{(x, y) \mid x \in X_{\text{train}} \subset X, y = \alpha(x) \in Y\}$ by applying sample elements $G_0 \subset G$ to the inputs. The new training data is then $D_{\text{train}}^{\text{aug}} := \{(g \cdot x, y) \mid (x, y) \in D_{\text{train}} \text{ and } g \in G_0\}$. A similar approach is to take a machine learning architecture $\beta$ and apply it to several $G$-translates of an input, before applying a pooling map to these different outputs. This yields a $G$-invariant map, and was studied in (2). A more sophisticated version of this pooling technique was proposed in (1).

We now turn to examples of *intrinsic* approaches. For neural networks, one can impose restrictions on the weights so that the resulting network is invariant under a group action on the input. This was done using group equivariant hidden layers, for example, in (18; 41). The same idea is also used in (29; 31; 32). Methods to determine all equivariant linear layers were proposed in (10; 15). Convolutional layers in neural networks are a standard tool to impose translational symmetry in image classification tasks. This idea has been generalised to group equivariant CNNs in (9) for actions by arbitrary discrete groups. Another intrinsic approach is proposed in (40, Section 2) based on the theory of *polynomial invariants* of $G$. All of these approaches are concerned with discrete symmetries. The study of continuous symmetries was initiated in (24) and expanded in (35; 11); and the case of Euclidean transformations has received additional attention, for example in (16; 37). An analogue of polynomial invariants that can handle some cases of continuous symmetries was proposed in (13).

## 1.2. Our contribution

Our approach to the problem is intrinsic, based on the fact that composing a $G$-invariant map with any other map, results in a $G$-invariant map. More precisely, we suggest a *G-invariant pre-processing step* to be applied to the input data that can then be composed with any machine learning architecture. The resulting composition is a $G$-invariant architecture. We provide a general framework to define the $G$-invariant preprocessing step for general group actions and a concrete implementation for finite groups acting by coordinate permutations.

One way of getting a $G$-invariant self-map of the feature space is to map to a so-called *fundamental domain* $\mathcal{F}$, which preserves the local geometry of the feature space. The set $\mathcal{F} \subset X$ comes with a $G$-invariant map $\pi \colon X \to \overline{\mathcal{F}}$ onto its closure. Let $\overline{\alpha}$ be the restriction of $\alpha$ to $\overline{\mathcal{F}}$, then by $G$-invariance $\alpha = \overline{\alpha} \circ \pi$. Instead of fitting a machine learning model $\beta \colon X \to Y$ to the training data $D_{\text{train}}$, we train the model $\overline{\beta} \colon \overline{\mathcal{F}} \to Y$ with $D_{\text{train}}^{\pi} := \{(\pi(x), y) \mid (x, y) \in D_{\text{train}}\} \subset \overline{\mathcal{F}} \times Y$ which approximates $\overline{\alpha}$. The resulting map $\beta = \overline{\beta} \circ \pi \colon X \to Y$ is $G$-invariant. Figure 1 shows the difference between the pre-processing approaches of augmentation and our method. This approach extends easily to $G$-*equivariant* machine learning, as explained at the end of Section 2.1.

More precisely, $G$-invariant maps from $X$ are parametrised by maps from the so-called *quotient space* $X/G$ of $X$, see Appendix A. The set $\overline{\mathcal{F}} \subset X$ locally models $X/G$ and, especially in the case $X = \mathbb{R}^n$ and $G$ acts by permuting coordinates, has the advantage of being extremely easy to compute. Another advantage our approach has is that it can be applied directly to any supervised machine learning model. In contrast, many existing methods, such as (18; 41; 29; 31; 32; 15; 10) only work for neural networks. The compu-

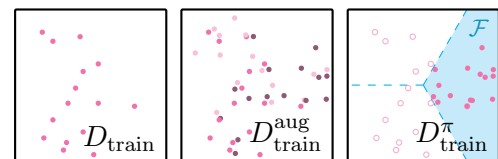

Figure 1: Example training data for a problem invariant under rotations by 120°; the training data after augmentation $D_{\text{train}}^{\text{aug}}$; and our approach $D_{\text{train}}^{\pi}$ (mapping all the data to the blue fundamental domain).

tational cost of data augmentation and many equivariant machine learning approaches scales with the size of the symmetry group. This is not the case for our approach, which we implement for groups of size $6 \cdot 10^{20}$ in Section 3.2. In the case of small group sizes, our approach makes no notable difference. That is for example the case in image recognition tasks with a rotational symmetry, which is one of our examples.

The rest of the paper is organised as follows. Section 2 describes our approach in detail, and compares it with other approaches from the literature. Section 3 discusses several applications of our approach, and compares the accuracies of machine learning architectures employing different approaches to $G$-invariant machine learning. Our main application is the learning of Hodge numbers from CICY matrices, first studied in (20), and our method improves on the state of the art (14).

## 2. Mathematical approach

In principle, our approach works in a very general setting, however we will restrict ourselves to the case that the feature space $X$ is a Riemannian manifold on which $G$ acts discretely by isometries. Recall an *isometry* is a map $X \to X$ which leaves the Riemannian metric unchanged. Given $x \in X$, its *orbit* under the action of $G$ is the set $G \cdot x = \{g \cdot x \mid g \in G\}$. Recall an action is discrete if every orbit under $G$ is a discrete subset of $X$. The reason for these restrictions is that the group action preserves the geometry of $X$.

### 2.1. Fundamental domains

We want to approximate a $G$-invariant function $\alpha$ satisfying Equation (1). It follows that $\alpha$ takes the same value on every element of any $G$-orbit. A set $R \subset X$ is a *set of orbit representatives* if for all $x \in X$, $R \cap (G \cdot x) \neq \emptyset$. Approximating $\alpha$ on a set of orbit representatives essentially approximates it everywhere. A nice choice for $R$ which takes into account the geometry of the group action is given by a fundamental domain.

**Definition 1** *Let a group $G$ act on $X$ discretely by isometries. A subset $\mathcal{F} \subset X$ is called a fundamental domain for $G$ if (a) it is open and connected; (b) every $G$-orbit intersects $\overline{\mathcal{F}}$, the closure of $\mathcal{F}$, in at least one point; and (c) whenever a $G$-orbit intersects $\overline{\mathcal{F}}$ at a point in $\mathcal{F}$, then this is the unique point of intersection with $\overline{\mathcal{F}}$.*

Given $G$ acting on $X$ we will find a $G$-invariant map $\pi\colon X \to \overline{\mathcal{F}}$, defined as $\pi(x) = \phi(x) \cdot x$, where $\phi\colon X \to G$ is some suitable function. We call such a map a *projection onto the fundamental domain $\mathcal{F}$*. We can now apply any machine leaning architecture to approximate the function $\alpha|_{\overline{\mathcal{F}}}\colon \overline{\mathcal{F}} \to Y$ trained on the data $D^\pi_{\text{train}}$ yielding a function $\overline{\beta}$. This can then be used to compute the $G$-invariant approximation for $\alpha$ defined on the whole of $X$ by defining $\beta = \overline{\beta} \circ \pi$, see Figure 2.

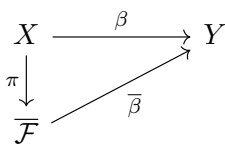

Figure 2: Defining $\beta$ from $\overline{\beta}$.

If $\overline{\beta}$ is a neural network, then the universal approximation property is satisfied, namely $\beta$ can approximate any continuous, $G$-invariant map $\alpha$ arbitrarily closely. This follows from the standard universal approximation theorem and is proved in Appendix E.1.

For generic $x \in X$ and any $g \in G$, $\phi$ has the property

$$\phi(g \cdot x) = \phi(x)g^{-1} \tag{2}$$

from which we can show that $\beta$ is indeed $G$-invariant:

$$\beta(g \cdot x) = \overline{\beta}(\phi(g \cdot x) \cdot (g \cdot x)) = \overline{\beta}((\phi(x)g^{-1}) \cdot (g \cdot x)) = \overline{\beta}((\phi(x) \cdot x) = \beta(x).$$

Our method of producing a $G$-invariant architecture can easily be modified to the $G$-equivariant setting. Let $G$ act on $X$ and $Y$, then a $G$-equivariant map $\alpha\colon X \to Y$ satisfies $\alpha(g \cdot x) = g \cdot \alpha(x)$ for all $x \in X$ and $g \in G$. Let $\pi\colon X \to \overline{\mathcal{F}}$ be the fundamental domain projection as above, and define $\phi\colon X \to G$ be a function such that $\pi(x) = \phi(x) \cdot x$. Then we define the $\beta$ model via

$$\beta(x) := \phi(x)^{-1} \cdot \overline{\beta}(\pi(x)) = \phi(x)^{-1} \cdot \overline{\beta}(\phi(x) \cdot x).$$

As above, for a generic $x \in X$, Equation (2) implies that $\beta$ is $G$-equivariant.

**Remark 2** *The projection $\pi$ is continuous on $\mathcal{F}$, but may fail to be continuous on the boundary of $\mathcal{F}$. Here $\phi$ does not necessarily satisfy Equation (2), and so the function $\beta$ may not be strictly $G$-invariant/equivariant. This only presents a problem if a significant portion of $D^\pi_{\text{train}}$ lies on the boundary. An example is in Section 3.*

Now we will describe two methods of finding a fundamental domain projection.

## 2.2. Dirichlet projections

The first method of computing a projection map follows from a classical proof of the existence of a fundamental domain for a discrete group $G$ acting by isometries on a metric space $(X, d)$ (every Riemannian manifold is automatically a metric space). The idea is to fix some point $x_0$ which is only fixed by elements of $G$ which fix the whole of $X$ point-wise, and define $\overline{\mathcal{F}}$ to be all points which are closer to $x_0$ than any other points in the orbit $G \cdot x_0$, see for example (36, §II.1.4). Such an $\mathcal{F}$ is called an *Dirichlet fundamental domain*. By its definition, we can rephrase the problem of finding a projection $\pi\colon X \to \overline{\mathcal{F}}$ as a minimisation problem for the metric on $X$: given $x \in X$ find $g \in G$ which minimises $d(g \cdot x, x_0)$. In practice, this can be approximated using a discrete gradient descent algorithm, see Appendix F.3. In the special case that $G$ acts on $\mathbb{R}^n$ by orthogonal matrices, equivalently one can minimise the inner product $\langle g \cdot x, x_0 \rangle$ instead of the distance.

### 2.3. Combinatorial projections

Finding a projection onto a Dirichlet fundamental domain may not always be the best option since the gradient descent algorithm must be applied to each input, and it cannot guarantee finding the global minimum, and hence can only be used to approximate $\pi$.

A preferable approach would be to have an explicit and easy to compute description of a projection $\pi$ onto a fundamental domain. We give an algorithm to compute such a projection for the rather general case that $G < S_n$ is a subgroup of the permutation group, which acts on $X = \mathbb{R}^n$ by permuting coordinates. More precisely, if $x = (x_i)_i \in \mathbb{R}^n$ and $s \in S_n$ we say $s$ acts on the left by $s \cdot (x_i)_i = \left(x_{s^{-1}(i)}\right)_i$. This induces an action of $G$ on $\mathbb{R}^n$ which we call a *permutation action of $G$*. This action is discrete (since $G$ is finite) and by isometries since it preserved the Euclidean metric.

**Example 3** *Let $G = S_n$ be the full permutation group acting on $\mathbb{R}^n$. Given a point $x = (x_1, \ldots, x_n) \in \mathbb{R}^n$ we can reorder the entries in any way by elements of $S_n$, and a fundamental domain corresponds to a consistent choice of reordering. One consistent choice would be to have the entries in increasing order, so $\mathcal{F} = \{(x_1, \ldots, x_n) \in \mathbb{R}^n \mid x_i < x_{i+1} \text{ for all } i < n\}$. The projection $\pi \colon \mathbb{R}^n \to \overline{\mathcal{F}}$ can be easily implemented using any sorting algorithm.*

*Another example of a simple group action is when $G = \mathbb{Z}_n < S_n$ acts by cyclically permuting the coordinates. In this case we can make a consistent choice by ensuring that the first entry is the smallest so $\mathcal{F} = \{(x_1, \ldots, x_n) \in \mathbb{R}^n \mid x_1 < x_i \text{ for all } i > 1\}$.*

The algorithm to find a fundamental domain projection in general is based on (12) in which the authors give an efficient algorithm to find a set of unique coset representatives for an arbitrary subgroup $G \leqslant S_n$, this is summarised in Appendix D.3. A set of coset representatives can be turned into a set of orbit representatives for the permutation action of $G$ on $\mathbb{R}^n$. Appendix D details how we modify their algorithm so that this set of orbit representatives is in fact a fundamental domain, and so that it outputs an explicit projection map. This map is easy to implement and efficient to compute. In fact, here we will define the *ascending projection $\pi_\uparrow$*, in Appendix B we discuss a few variations of this projection map and compute an example. We have listed the outputs of the algorithm for several common examples of groups $G \leqslant S_n$ in Appendix C.

**Finding a base** Let $N = \{1, \ldots, n\}$, which we identify with the set of indices for the standard basis for $\mathbb{R}^n$ so $S_n$ acting on $\mathbb{R}^n$ corresponds to the right action of $S_n$ on $N$ by $i \cdot s = s^{-1}(i)$. The first step of the algorithm is to find a *base* for $G \leqslant S_n$.

**Definition 4** *A base for $G \leqslant S_n$ is an ordered subset $B = (b_1, \ldots, b_k)$ of $N$ such that $\bigcap_{i=1}^k \operatorname{Stab}_G(b_i) = \{1\}$, where $\operatorname{Stab}_G(b_i) = \{g \in G \mid b_i \cdot g = b_i\}$ is the* stabiliser *of $b_i$ in $G$. Given a base let $G_0 = G$ and for $1 \leqslant i \leqslant k$, define $G_i = \operatorname{Stab}_{G_{i-1}}(b_i) = G_{i-1} \cap \operatorname{Stab}_G(b_i)$.*

It follows that $G_k = \{1\}$. One can always choose $B = (1, \ldots, n-1)$ as a base, although an algorithm to compute an efficient base is given in Appendix D.3. Given a base $B$ and the groups $G_i$, we will also define $\Delta_i$ to be the orbit of $b_i$ under the action of $G_{i-1}$.

**Example 5** *Let $G$ be the subgroup in $S_4$ generated by the elements $(1\ 2)$ and $(3\ 4)$, where we represent permutations using cycle notation: eg $(1\ 2)$ swaps $1$ and $2$ and fixes $3$ and $4$. Then $B = (1, 3)$ is a base and we have stabilisers $G_0 = \{e, (1\ 2), (3\ 4), (1\ 2)(3\ 4)\}$, $G_1 = \{e, (3\ 4)\}$, $G_2 = \{e\}$, and orbits $\Delta_1 = \{1, 2\}$, $\Delta_2 = \{3, 4\}$.*

**Perturbing points in $\mathbb{R}^n$**  We now need to define the map $\phi_\uparrow \colon X \to G$ used in the definition of $\pi_\uparrow$. The map $\phi_\uparrow$ will only be uniquely defined on points $x = (x_i)_i \in \mathbb{R}^n$ all of whose entries are distinct. We first perturb $x$ slightly to get a point with this property. Choose a *perturbation vector* $\varepsilon$ which has all distinct entries, for example $\varepsilon = \frac{1}{2n}(1, 2, \ldots, n)$. Let $d = \min_{x_i \neq x_j}\{|x_i - x_j|\}$ (choose $d = 1$ if all entries of $x$ are the same) and define $x' = x + d\varepsilon$, which is guaranteed to have all entries distinct. The entries of $x'$ have the same relative order, ie if $x'_i \leqslant x'_j$ then $x_i \leqslant x_j$, and $\phi_\uparrow$ will depend only on this relative ordering of entries. Then we define $\phi_\uparrow(x) = \phi_\uparrow(x')$ where $\phi_\uparrow(x')$ is defined below.

**The ascending projection map**  We will define a sequence of permutations $g_i \in G$ for $1 \leqslant i \leqslant k$ as follows. Assume $g_1, \ldots, g_{i-1}$ have already been found. $G_{i-1}$ acts transitively on $\Delta_i$, choose $j \in \Delta_i$ such that the $j$th entry of $(g_{i-1} \cdots g_1) \cdot x'$ is minimal among those entries indexed by $\Delta_i$. Choose $g_i \in G_{i-1}$ such that $j \cdot g_i = g_i^{-1}(j) = b_i$. Now define $\phi_\uparrow(x') \coloneqq g_k \cdots g_1$, note the choice of the $g_i$'s is not unique, but we will show in Appendix D.6 that $\phi(x')$ *is* uniquely defined. Appendix D is devoted to the proof of the following theorem which says that the map we have defined is a projection onto a fundamental domain.

**Theorem 6**  *Define $\pi_\uparrow \colon \mathbb{R}^n \to \mathbb{R}^n$ by $\pi_\uparrow(x) = \phi_\uparrow(x) \cdot x$, and let $\mathcal{F}$ be the interior of its image. Then $\mathcal{F}$ is a fundamental domain for $G$ acting on $\mathbb{R}^n$. Given a choice of base $B$ and perturbation vector $\varepsilon$, the projection $\pi_\uparrow$ is uniquely defined.*

**Example 3 (Continued)**  *For $G = S_n$, then $B = (1, 2, \ldots, n-1)$ is the smallest base we can use, so $G_i = \mathrm{Perm}(i+1, \ldots, n)$ and $\Delta_i = \{i, \ldots, n\}$. Fixing $x' \in X$ and following the algorithm above: $g_1 \in G_0 = S_n$ is a permutation moving the smallest entry indexed by $\Delta_1 = \{1, \ldots, n\}$ to the entry indexed by $b_1 = 1$. Repeating this for each $i$ up to $n-1$, $g_i$ moves the $i$th smallest entry of $x'$ to the $i$th position. The result is that $(g_{n-1} \cdots g_1) \cdot x'$ has entries ordered from smallest to largest.*

*In a very similar way, applying the algorithm to $G = \mathbb{Z}_n$ using the base $B = (1)$ yields the same projection onto a fundamental domain described at the start of Section 2.3.*

In Appendix F.2 we discuss an algorithm to compute $\pi_\uparrow$ for arbitrary permutation groups, and analyse the time and space complexity of this algorithm. The worse case runtime of the algorithm depends only on the dimension $n$ and the size of the base $k < n$, so in particular the algorithm remains computationally tractable for very large groups. In general, the time complexity is $O(k^2 n^3)$ (Theorem 27). For many groups this drops to $O(n^4 (\log \log n)^2)$ (Theorem 33), and for the common groups discussed in Appendix C, it drops to $O(n^2)$.

### 2.4. Comparing approaches to invariant and equivariant machine learning

We can compare the various approaches to invariant machine learning discussed in Section 1.1 on a theoretical level; below we compare them experimentally. Augmentation is a data pre-processing step and can be applied to any model. However, the resulting model need not be $G$-invariant, and for large groups it is computationally impractical to augment by a representative subset of the group.

As for intrinsic approaches, group equivariant neural networks like (9; 15; 29; 41) are model-specific, and there are unavoidable limits on the universality while using low-order tensors (28). Additionally, (9) requires the elements of $G$ to be stored in memory, making

it impractical for large groups. The approach in (40) using polynomial invariants. The approach in (40) is not practical outside of small group actions owing, in part, to the need to compute a basis of polynomial invariants. This basis will in general be large, increasing the dimension of the feature space dramatically.

Fundamental domain projections are computationally easy to use, maintain the original dimension and geometry of the data, and are compatible with any machine learning model. The resulting model is $G$-invariant almost everywhere (see Remark 2).

## 3. Examples and results

We show how $G$-invariant pre-processing can be applied to examples of classification tasks in group theory, string theory, and image recognition. In each case, the symmetry group acts differently on the input space. We account for this by choosing appropriate projection maps from the previous section. Experiments 3.1 and 3.3 are chosen as proof of concept, not to reach the state of the art, the main application of our approach is experiment 3.2. Implementation details may be found in Appendix F.1.

### 3.1. Cayley tables

Multiplication tables of groups are called *Cayley tables*. The following model problem was introduced in (21, Section 3.2.3): up to isomorphism, there are 5 groups with 8 elements. Separate their Cayley tables into two classes and apply random permutations until $20\,000$ tables in each class exist. The problem is to assign a given table to one of two classes. This map is invariant under the action of $S_8 \times S_8$ acting on $\mathbb{R}^{8 \times 8}$ by row and column permutations.

Let $\pi_\uparrow \colon \mathbb{R}^{8 \times 8} \to \mathbb{R}^{8 \times 8}$ be the ascending projection map from Section 2.3, in particular as defined in Appendices C.1 and C.6. This has an explicit description as follows: given a choice of total order on the group elements, permute the columns so that the first row is ordered smallest to biggest, and then permute the rows so that the first column is ordered smallest to biggest. Then, $\pi_\uparrow$ is invariant under the action of $S_8 \times S_8$ and can be efficiently computed for Cayley tables. This pre-processing effectively 'undoes' the permutations, which makes the machine learning problem trivial. Consequently, we achieve nearly perfect accuracy using a linear support vector machine (SVM), see Table 1.

We compare our approach with the neural network from (21), with the Deep Sets architecture from (41), and with the $S_8 \times S_8$-invariant neural network from (18). The Deep Sets architecture is invariant under the action of the full $S_{8\cdot8} = S_{64}$ on $\mathbb{R}^{8 \times 8}$. As all Cayley tables are in the same orbit under this group action, the performance of this architecture can only be as good as random guessing. Note that the general purpose architectures described in (29; 15) in this case are identical to (18). Other architectures from the literature, such as (9), are difficult to apply to this problem, since they require keeping a non-sparse map $S_8 \times S_8 \to \mathbb{R}$ in memory. This group has size $8! \cdot 8! \approx 1.6 \cdot 10^9$.

### 3.2. CICY

Hodge numbers are crucial for understanding and distinguishing Calabi-Yau manifolds. Algebraic algorithms to compute them explicitly exist, and in (17), a dataset of complex three-dimensional *complete intersection Calabi-Yau* manifolds (CICYs) and their Hodge

Table 1: Accuracy of predicting the group isomorphism type of a Cayley table.

|  | Accuracy |
| --- | --- |
| MLP (21) | $0.501 \pm 0.015$ |
| Deep Sets (41) | $0.504 \pm 0.010$ |
| $G$-inv MLP (18; 29; 15) | $0.498 \pm 0.012$ |
| $\pi_\uparrow$+SVM | $\mathbf{0.994 \pm 0.008}$ |

numbers is given. Hodge numbers are expensive to compute, and it is not feasible to compute all Hodge numbers explicitly on larger datasets. For applications in string theory, it is important to have Calabi-Yau manifolds with large first Hodge number. In (20), it was suggested to use a neural network to *approximately* compute (among other tasks) the first Hodge number of a Calabi-Yau manifold, thereby quickly identifying the most promising candidates without resorting to the expensive exact calculation.

For three-dimensional CICYs, the Hodge numbers have been computed, but it remains the most commonly used benchmark for comparison of machine learning models in the field, so we apply our algorithm to this problem. Here, CICYs are represented by matrices of size up to $12 \times 15$, and the first Hodge number is an integer. The same problem was subsequently studied in (5; 6; 14), using more sophisticated machine learning models. The problem is invariant under row and column permutations, ie an action of $S_{12} \times S_{15}$ on $\mathbb{R}^{12 \times 15}$, but none of the machine learning models which have been implemented previously for the Hodge number classification satisfy this invariance.

We compare two pre-processing maps: the map $\pi_{\mathrm{Dir}} \colon \mathbb{R}^{12 \times 15} \to \mathbb{R}^{12 \times 15}$ defined in Section 2.2, which we computed by performing discrete gradient descent; and $\pi_\uparrow$ defined in the same way as in Section 3.1. We found that composing $\pi_{\mathrm{Dir}}$ with existing neural networks slightly improves performance, but not significantly. We also considered an alternative training task in which input matrices first had their rows and columns randomly permuted. In this case, our model outperforms models from the literature by a large margin. We also compare our model with the group invariant model from (18) in both training tasks, see Table 2. Again, the approaches of (29) and (15) reduce to (18). As for Cayley tables, the approach in (9) is impractical due to the large group size $12! \cdot 15! \approx 6 \cdot 10^{20}$.

As our approach is intrinsic it is well suited for problems with a large symmetry group. For all networks but the $G$-invariant multi layer perceptron (MLP) the accuracy decreases on the permuted dataset. This suggests that the rows and columns of the CICY matrices are already systematically ordered in the original dataset. The map $\pi_\uparrow$ can be computed efficiently but need not be $G$-invariant on the boundary of the fundamental domain by Remark 2. This is a potential problem since the input data, which consists of integer-valued matrices, is discrete. Indeed, a substantial proportion of the CICY matrices are very sparse and do lie on the boundary, which could be the reason why $\pi_\uparrow$ performs relatively poorly on the permuted data set. The projection map $\pi_{\mathrm{Dir}}$ can only be approximated but is fully $G$-invariant which is a crucial advantage on the permuted dataset.

Table 2: Accuracies for the task of predicting the second Hodge number of a CICY matrix. Models are compared on the original training task and on randomly permuted input matrices. The last three rows are group invariant models, the first three rows are not group invariant models.

|  | Original dataset | Randomly permuted |
|---|---|---|
| MLP (20) | $0.554 \pm 0.015$ | $0.395 \pm 0.029$ |
| MLP+pre-processing (6) | $0.858 \pm 0.009$ | $0.417 \pm 0.086$ |
| Inception (14) | $0.970 \pm 0.009$ | $0.844 \pm 0.117$ |
| $G$-inv MLP (18; 29; 15) | $0.895 \pm 0.029$ | $0.914 \pm 0.023$ |
| $\pi_{\mathrm{Dir}}$+Inception | $\mathbf{0.975 \pm 0.007}$ | $\mathbf{0.963 \pm 0.016}$ |
| $\pi_{\uparrow}$+Inception | $0.969 \pm 0.009$ | $0.539 \pm 0.020$ |

### 3.3. Classifying rotated handwritten digits

As an instructional example, we use the MNIST dataset of handwritten images from (26), on which $\mathbb{Z}_4$ rotates the images by multiples of $90°$. We use the ascending averaging projection defined in Appendix B, $\pi_{\uparrow\mathrm{av}} \colon \mathbb{R}^{28 \times 28} \to \mathbb{R}^{28 \times 28}$. This map rotates each image so that its brightest quadrant is at the top-left. We then compare performance of a linear classifier, a shallow neural network, and SimpNet (see (19)) which is among the top performers on the original MNIST task; first on their own, then with data augmentation, and finally with the projection map $\pi_{\uparrow\mathrm{av}}$, but without data augmentation, see Table 3.

Table 3: Accuracy for the task of recognising handwritten digits. We use two different degrees of data augmentation: either add every possible rotation of the input image to the training data ($Aug. \times 4$) or applying data augmentation until the training data reaches 1.5 times its original size ($Aug. \times 1.5$).

|  | No pre-processing | Aug. $\times 1.5$ | Aug. $\times 4$ | $\pi_{\uparrow\mathrm{av}}$ |
|---|---|---|---|---|
| Linear | $0.677 \pm 0.001$ | $0.682 \pm 0.001$ | $0.682 \pm 0.001$ | $0.784 \pm 0.001$ |
| MLP | $0.939 \pm 0.001$ | $0.963 \pm 0.002$ | $0.963 \pm 0.001$ | $0.953 \pm 0.003$ |
| SimpNet (19) | $0.979$ | $0.986$ | $0.986$ | $0.979$ |

For linear classifiers, data augmentation does not improve accuracy substantially due to their small number of parameters. Unsurprisingly, pre-processing with $\pi_{\uparrow\mathrm{av}}$ improves performance because it is partially successful at rotating pictures into a canonical orientation.

For neural networks with more than one layer, data augmentation increases accuracy, because the model now has sufficient parameters to include the information from the additional training data. Pre-processing using $\pi_{\uparrow\mathrm{av}}$ yields better accuracy than no pre-processing, but

worse accuracy than full data augmentation. If fewer training data points are added during the data augmentation step, the benefit is comparable to applying the map $\pi_{\uparrow \mathrm{av}}$.

This is one example of the fact that data augmentation may be the best pre-processing option if the symmetry group $G$ has few elements and one can augment by the full group. If $|G|$ is very large, this is not possible, and pre-processing using a fundamental domain projection may be better than augmenting with a small, non-representative, subset of $G$.

## 4. Conclusion

The $G$-invariant pre-processing step proposed in this paper has a clear mathematical motivation. It respects the geometry of the input space and we show in Appendix A it naturally fits into a larger framework of $G$-equivariant machine learning. There are also many practical advantages of our approach: it can be applied to any machine learning architecture, it preserves the dimension of the input space and in most cases it guarantees perfect $G$-invariance. Furthermore, the computation cost is generally low even if $|G|$ is very large. For the image recognition task where $|G| = 4$, many of these advantages are only relevant for networks with a small number of neurons. For Cayley tables however, $|G| = 8! \cdot 8! \approx 1.6 \cdot 10^9$ is very large and the $G$-invariance of our approach produces nearly perfect accuracy. The symmetry group is even larger for CICY matrices even though both the action of the symmetry group as well as the classification task itself are more complicated. Our approach significantly improves the most accurate architecture known so far for this task.

While our experiments all fall into the category of finite groups acting by permutations on $\mathbb{R}^n$, our approach works in the much broader setting of (possibly infinite) groups acting discretely by isometries on manifolds. Although not discussed in the paper, it also generalises to continuous Lie group actions, with the fundamental domain replaced by a geometrically nice set of orbit representatives.

## Acknowledgments

We would like to thank Yang-Hui He and Momchil Konstantinov for their comments on a draft of this paper. We would also like to thank the anonymous reviewers for their helpful comments. All three authors were supported by the Engineering and Physical Sciences Research Council [EP/L015234/1]. The EPSRC Centre for Doctoral Training in Geometry and Number Theory (The London School of Geometry and Number Theory), University College London. The second author was supported by the Simons Collaboration "Special Holonomy in Geometry, Analysis, and Physics" and the first author by the London Mathematical Society during part of the work.

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

## Appendix A. Unifying intrinsic approaches to equivariant machine learning

In Sections 2 and 3 we approximated $G$-invariant functions by considering them as functions on a fundamental domain. In this appendix we show that approximating a $G$-invariant function is equivalent to approximating it on the *quotient space*. We then pick up two approaches to $G$-invariant machine learning from the literature alongside our approach, and explain in which sense they can be viewed as machine learning on quotient spaces. It will be convenient to treat the $G$-invariant problem as a special case of the $G$-equivariant problem, since an invariant function $X \to Y$ is an equivariant function where $G$ acts trivially on $Y$. We return to the setting that $G$ acts by isometries on $X$ and $Y$ which are Riemannian manifolds. To simplify the proof of Theorem 9 we make the stronger assumption that the action is properly discontinuous, rather than discrete, which means that for any compact subset $K$, the set $\{g \in G \mid g \cdot K \cap K \neq \emptyset\}$ is finite.

### A.1. Universality of quotient spaces

We begin by defining the quotient space of a group action, and discuss its properties.

**Definition 7**  *Let $G$ be a group acting on $X$, then the* quotient space $X/G$ *is the set of all $G$-orbits of points in $X$, $\{G \cdot x \mid x \in X\}$. A quotient space is automatically equipped with a $G$-invariant map $\pi_X \colon X \to X/G \colon x \mapsto G \cdot x$. If $X$ is a subset of $\mathbb{R}^n$ and the action is by isometries, $X/G$ inherits a metric from the Riemannian metric on $X$, and $\pi_X$ is a local isometry.*

For us the key is that quotient spaces are *universal* with respect to $G$-equivariant maps.

**Universal property of quotient spaces**   Given two spaces $X$ and $Y$ and a group $G$ acting on both, let $\pi_X \colon X \to X/G$ and $\pi_Y \colon Y \to Y/G$ be the canonical projection maps. Then for any $G$-equivariant map $\alpha \colon X \to Y$ there is a unique map $\overline{\alpha} \colon X/G \to Y/G$ such that $\pi_Y \circ \alpha = \overline{\alpha} \circ \pi_X$ (ie the diagram on the right commutes).

$$
\begin{array}{ccc}
X & \xrightarrow{\;\;\alpha\;\;} & Y \\
\downarrow{\scriptstyle \pi_X} & & \downarrow{\scriptstyle \pi_Y} \\
X/G & \dashrightarrow{\;\overline{\alpha}\;} & Y/G
\end{array}
$$

If the action of $G$ on $X$ and $Y$ is sufficiently 'nice', then the converse holds: equivariant maps $X \to Y$ are parametrised by certain maps $X/G \to Y/G$. Notice that for any $G$-equivariant function $\alpha$, point stabilisers in $X$ and $Y$ have the property that for any $x \in X$, $\mathrm{Stab}_G(x) \subset \mathrm{Stab}_G(\alpha(x))$.

**Definition 8**  *Let $X$ and $Y$ be simply connected, a continuous map $\overline{\alpha} \colon X/G \to Y/G$ is* compatible *with the $G$ actions if for any $x \in X$, the stabiliser $\mathrm{Stab}_G(x)$ is conjugate in $G$ to a subgroup of $\mathrm{Stab}_G(y)$, where $y$ is a lift under $\pi_Y$ of $\overline{\alpha}(\pi_X(x))$.*

It follows from the observation above that if $\overline{\alpha}$ is the map coming from the universal property, then $\overline{\alpha}$ is automatically compatible with the $G$-action. In the special case that $G$ acts trivially on $Y$ (ie the $G$-invariant case) then every continuous map $\overline{\alpha} \colon X/G \to Y/G = Y$ is compatible.

**Theorem 9** *Any compatible function $\overline{\alpha}\colon X/G \to Y/G$ lifts to a $G$-equivariant function $\alpha\colon X \to Y$. Suppose $\alpha'$ is another such lift, and assume there is some $x_0 \in X$ such that $\alpha'(x_0) = \alpha(x_0)$ where $\mathrm{Stab}_G(\alpha(x_0))$ fixes $Y$ point-wise. Then $\alpha'(x) = \alpha(x)$ for all $x \in X$.*

We omit the proof, which can be deduced using (8, Theorem 4.1.6). This gives a converse to the universal property, and shows that up to an isometry of $Y$ there is a one-to-one correspondence between $G$ equivariant maps $X \to Y$, and compatible maps of their quotients.

Now suppose $\alpha\colon X \to Y$ is a $G$-equivariant function we want to approximate using a supervised machine learning algorithm. Using an intrinsic approach, as discussed in Section 1.1, means approximating $\alpha$ by a function $\beta$ which is *a priori* $G$-equivariant. By the theorem above, this is equivalent to approximating $\overline{\alpha}$ by a compatible function $\overline{\beta}$. Below we will discuss how different intrinsic approaches to the equivariant machine learning problem fit into this framework.

## A.2. Equivariant maps from polynomial invariants

We discuss briefly the approach proposed by Yarotsky in (40, Section 2) based on the theory of polynomial invariants of a group $G$ acting on $X = \mathbb{R}^n$. A polynomial $p(x) \in \mathbb{R}[x_1, \ldots, x_n]$ is called a *polynomial invariant* if it satisfies Equation (1). Similarly, if $G$ also acts on $\mathbb{R}^m$, then $q(x)\colon \mathbb{R}^n \to \mathbb{R}^m$ is a *polynomial equivariant* if $q(g \cdot x) = g \cdot q(x)$ for all $g \in G$ and $x \in \mathbb{R}^n$.

The following, proved in (30) in the invariant case, was generalised to compact Lie groups in (38, Theorem 8.14.A). The equivariant case is proved in (39, Section 4). If a finite group $G$ acts on $\mathbb{R}^n$ and $\mathbb{R}^m$, then there are finite sets of invariants $\{p_i(x)\}_{i=1}^k$, and equivariants $\{q_j(x)\}_{j=1}^l$ such that any polynomial equivariant $q(x)$ can be written as $q(x) = \sum_{j=1}^l q_j(x) r_j(p_1(x), \ldots, p_k(x))$ for some $r_j(x) \in \mathbb{R}[x_1, \ldots, x_k]$. In the invariant case we can take $l = 1$, and $q_1(x) = 1$.

Yarotsky shows in (40, Proposition 2.4) that any continuous $G$-equivariant function $\alpha\colon \mathbb{R}^n \to \mathbb{R}^m$ can be approximated on a compact set by a $G$-equivariant neural network of the form

$$\beta(x) = \sum_{j=1}^l q_j(x) \sum_{h=1}^d a_{jh}\, \sigma\left(\sum_{i=1}^k b_{jhi} p_i(x) + c_{jh}\right) \tag{3}$$

for some $a_{jh}, b_{jhi}, c_{jh} \in \mathbb{R}$, where $d \in \mathbb{N}$, and $\sigma$ is a continuous non-polynomial activation function.

Notice that all the learnable parameters are contained in the inner two sums, which also constitute the neural network in the $G$-invariant case $\sum_{h=1}^d a_h\, \sigma\left(\sum_{i=1}^k b_{hi} p_i(x) + c_h\right)$, see (40, Proposition 2.3). The relationship between this method and quotient spaces is shown by the following result (33).

**Theorem 10** *The map $p(x) := (p_1(x), \ldots, p_k(x))$ factors through $\mathbb{R}^n/G$, and induces a smooth embedding of $\mathbb{R}^n/G$ into $\mathbb{R}^k$.*

We can reinterpret the invariant version of Equation (3) as a fully-connected neural network $\overline{\beta}$. We then train this network on the data $D_{\text{train}}^p = \{(p(x), y) \mid (x, y) \in D_{\text{train}}\}$ which has been projected to the quotient space by $p$. Thus $\overline{\beta}$ learns the map $\overline{\alpha}$ directly.

### A.3. Our approaches

Our approach of projecting onto a fundamental domain (whether by a combinatorial projection or a Dirichlet projection) fits very naturally in this general framework. Like Yarotsky's approach, we want to try to approximate the function $\overline{\alpha}$ rather than approximating $\alpha$. Instead of working directly with the quotient spaces, one can think of the map from the fundamental domain to the quotient space $\pi_X|_{\mathcal{F}_X} : \mathcal{F}_X \to X/G$ as a *chart* in the sense of differential geometry, and so $\mathcal{F}_X$ locally parametrises $X/G$.

In the invariant case, we can approximate $\alpha = \overline{\alpha} \circ \pi_X|_{\mathcal{F}_X}$ by approximating $\overline{\alpha}$. In the equivariant case, we can also view $\pi_Y|_{\mathcal{F}_Y} : \mathcal{F}_Y \to Y/G$ as a chart, and because $\pi_Y|_{\mathcal{F}_Y}$ is a bijection onto its image we can apply its inverse and approximate $\alpha = (\pi_Y|_{\mathcal{F}_Y})^{-1} \circ \overline{\alpha} \circ \pi_X|_{\mathcal{F}_X}$ by approximating $\overline{\alpha}$. Note that $\pi_X|_{\mathcal{F}_X}$ and $\pi_Y|_{\mathcal{F}_Y}$ are not surjective in general, and there is no canonical way to extend their domain to make them so. The fix for this is to perturb points to lie in the preimage of $\mathcal{F}_X$ as discussed in Section 2.3.

**Remark 11** *We do not work directly with the quotient spaces in our approach because the quotient space of a vector space is not itself a vector space. It needs to be embedded first in another vector space before being used as the input for a neural network, say. Finding quotient space embeddings is an extremely difficult problem.*

*The approach in (40) is to use polynomial invariants, which can be found using an algorithm. The problem, in addition to being computationally infeasible in practice (the target dimension in the case of $28 \times 28$ pixel images as in Section 3.3 would be of order $10^8$), is that they significantly distort the training data leading to low accuracy.*

*To avoid this one must find an* isometric *embedding which does not distort the data. However, this is even more difficult, and likewise significantly increases the ambient dimension of the training data. Further details on both the distortion of training data by polynomial invariants and quotient space projections may be found in (34).*

### A.4. Equivariant layers in neural networks

On the face of it, the various approaches to equivariant neural networks such as (29; 9; 15) bypass the compatible map $\overline{\alpha}$ by approximation $\alpha$ directly, restricting the space of maps which can be used. The central problem in these approaches is computing what the possible equivariant maps are, and (15) gives a very general approach to this problem. The unified approach we discuss above provides another possible approach to this problem based on geometric methods as opposed to representation theory. We sketch this in Appendix E.2.

## Appendix B. Other combinatorial projection maps

There are three natural variations of the combinatorial projection map $\pi_\uparrow$ we defined in Section 2.3 which may be more suited to specific applications. We called that projection an *ascending projection*. The variations are a *descending projection* $\pi_\downarrow$, and ascending and descending *averaging projections* $\pi_{\uparrow\text{av}}$ and $\pi_{\downarrow\text{av}}$. These projections each have their own version of Theorem 6 whose proof is essentially identical.

The descending projection is defined via $\phi_\downarrow$, which differs from $\phi_\uparrow$ only when we define $g_i$. In this case $G_{i-1}$ acts transitively on $\Delta_i$, and we choose $j \in \Delta_i$ such that the $j$th entry

of $(g_{i-1} \cdots g_1) \cdot \hat{x}$ is *maximal* among those entries indexed by $\Delta_i$. Choose $g_i \in G_{i-1}$ such that $j \cdot g_i = g_i^{-1}(j) = b_i$. If the input data for the machine learning algorithm consisted of vectors containing non-negative entries including many zeros, the descending projection in some sense *prioritises* the non-zero entries, so may yield different results.

For the averaging projections, assume that $G = H_1 \times H_2$ is a direct product of groups $H_j \leqslant S_{n_j}$ which acts the space of $n_1 \times n_2$ matrices, $\mathbb{R}^{n_1} \otimes \mathbb{R}^{n_2}$, by letting $H_1$ permute rows and $H_2$ permute columns. In this case, identify $N$ with the set of pairs

$$\{(l, m) \mid 1 \leqslant l \leqslant n_1, \ 1 \leqslant m \leqslant n_2\}.$$

Define a transformation $\mu \colon \mathbb{R}^{n_1} \otimes \mathbb{R}^{n_2} \to \mathbb{R}^{n_1} \otimes \mathbb{R}^{n_2}$ by

$$\mu \colon (x_{lm})_{lm} \mapsto \left( \frac{1}{n_1}(x_{1m} + x_{2m} + \cdots + x_{n_1 m}) + \frac{1}{n_2}(x_{l1} + x_{l2} + \cdots + x_{l n_2}) \right)_{lm}.$$

Notice this is a $G$-equivariant linear map which replaces each entry of $(x_{lm})_{lm}$ by the sum of the averages of the entries in its row and column. Now for any $x \in \mathbb{R}^{n_1} \otimes \mathbb{R}^{n_2}$ we define

$$\phi_{\uparrow \mathrm{av}}(x) = \phi_\uparrow \left( \widehat{\mu(x)} \right) \quad \text{and} \quad \phi_{\downarrow \mathrm{av}}(x) = \phi_\downarrow \left( \widehat{\mu(x)} \right).$$

These definitions generalise in the obvious way to the case $G = \prod_{j=1}^r H_j$ acting on $\bigotimes_{j=1}^r \mathbb{R}^{n_j}$ component-wise, where $H_j \leqslant S_{n_j}$. One might wish to use an averaging projection if, for example, one of the $H_j$'s is trivial, in which case a non-averaging projection ignores most of the entries, since they will not be in any of the orbits $\Delta_i$. This is the case in the application discussed in Section 3.3.

**Example 12** *Let $G = \mathbb{Z}_3 \times S_3 \leqslant S_3 \times S_3$ act on $\mathbb{R}^3 \otimes \mathbb{R}^3$, thought of as the set of $3 \times 3$ matrices, by cyclically permuting the rows and freely permuting the columns. In this case let $N = \{(l, m) \mid 1 \leqslant l \leqslant 3, \ 1 \leqslant m \leqslant 3\}$ and construct a base. Let $b_1 = (1, 1)$ whose stabiliser is $G_1 = \{1\} \times \mathrm{Sym}(\{2, 3\})$, and the orbit of $b_1$ under $G_0 = G$ is $\Delta_1 = N$. Now $(2, 1)$ and $(3, 1)$ are both fixed by $G_1$ and so should not be the next element of the base. Choose $b_2 = (1, 2)$. Then $G_2 = \{1\} \times \{1\}$ and the orbit of $b_2$ under $G_1$ is $\Delta_2 = \{(1, 2), (1, 3)\}$. Since $G_2 \cong \{1\}$ we are done and $B = ((1, 1), (1, 2))$.*

*Let $x' = (x'_{lm})_{lm}$ be a $3 \times 3$ matrix whose entries are distinct, we want to compute $\phi_\uparrow(x')$. Let $(p_1, q_1) \in \Delta_1 = N$ be the pair such that $x'_{p_1 q_1} = 1$ is the minimal entry in $x'$. Then we can choose $g_1 = (s_1, (1 \ q_1)) \in \mathbb{Z}_3 \times S_3$ where*

$$s_1 = \begin{cases} (1) & p_1 = 1 \\ (1\ 2\ 3) & p_1 = 2 \quad \in \mathbb{Z}_3 \\ (1\ 3\ 2) & p_1 = 3 \end{cases}$$

*Now let $g_1 \cdot x' = (x''_{lm})_{lm}$, and let $(1, q_2) \in \Delta_2 = \{(1, 2), (1, 3)\}$ minimise $x''_{1 q_2}$. Define $g_2 = ((1), (2 \ q_2)) \in G_1$ and $\phi_\uparrow(x') = g_2 g_1$.*

*Combinatorially we can describe the projection $\pi_\uparrow$ as follows: transport the smallest entry of $x'$ to the top left corner by cyclically permuting rows and freely permuting columns, and then order columns 2 and 3 so that the entries in the first row increase.*

*As an example, consider the matrix $x$ and perturbation matrix $\varepsilon$*

$$x = \begin{pmatrix} 5 & 3 & 3 \\ 4 & 0 & 0 \\ 3 & 5 & 1 \end{pmatrix}, \quad \varepsilon = \frac{1}{18}\begin{pmatrix} 1 & 2 & 3 \\ 4 & 5 & 6 \\ 7 & 8 & 9 \end{pmatrix} \implies x' = x + \varepsilon = \frac{1}{18}\begin{pmatrix} 91 & 56 & 57 \\ 76 & 5 & 6 \\ 61 & 98 & 27 \end{pmatrix}.$$

*We can now apply $\pi_\uparrow(x) = \phi_\uparrow(x') \cdot x$ in the two step process described above:*

$$x' = \frac{1}{18}\begin{pmatrix} 91 & 56 & 57 \\ 76 & 5 & 6 \\ 61 & 98 & 27 \end{pmatrix} \xmapsto{g_1} \frac{1}{18}\begin{pmatrix} 5 & 76 & 6 \\ 98 & 61 & 27 \\ 56 & 91 & 57 \end{pmatrix} \xmapsto{g_2} \frac{1}{18}\begin{pmatrix} 5 & 6 & 76 \\ 98 & 27 & 61 \\ 56 & 57 & 91 \end{pmatrix} = \pi_\uparrow(x')$$

$$x = \begin{pmatrix} 5 & 3 & 3 \\ 4 & 0 & 0 \\ 3 & 5 & 1 \end{pmatrix} \xmapsto{g_1} \begin{pmatrix} 0 & 4 & 0 \\ 5 & 3 & 1 \\ 3 & 5 & 3 \end{pmatrix} \xmapsto{g_2} \begin{pmatrix} 0 & 0 & 4 \\ 5 & 1 & 3 \\ 3 & 3 & 5 \end{pmatrix} = \pi_\uparrow(x).$$

*Similarly*

$$\pi_\downarrow(x) = \begin{pmatrix} 5 & 3 & 1 \\ 3 & 5 & 3 \\ 0 & 4 & 0 \end{pmatrix}.$$

*We can also compute the averaging versions of these projections*

$$\mu(x) = \frac{1}{3}\begin{pmatrix} 23 & 19 & 15 \\ 16 & 12 & 8 \\ 21 & 17 & 13 \end{pmatrix} \implies \pi_{\uparrow\mathrm{av}}(x) = \begin{pmatrix} 0 & 0 & 4 \\ 1 & 5 & 3 \\ 3 & 3 & 5 \end{pmatrix}, \text{ and } \pi_{\downarrow\mathrm{av}}(x) = \begin{pmatrix} 5 & 3 & 3 \\ 4 & 0 & 0 \\ 3 & 5 & 1 \end{pmatrix}.$$

## Appendix C. Examples of combinatorial projection maps

In this section we list combinatorial projection maps for several common examples of groups $G \leqslant S_n$. Notice that in each of the four examples of concrete groups below, implementation via a suitable sorting function circumvents the need to perturb inputs initially.

### C.1. The symmetric group

If $G = S_n$, let $N = \{1, \ldots, n\}$ and we can choose the base $B = (1, 2, \ldots, n-1)$. The ascending projection $\pi_\uparrow(x)$ permutes the entries so that they increase from left-to-right, and the descending projection $\pi_\downarrow(x)$ permutes the entries so that they decrease.

### C.2. The alternating group

If $G = A_n < S_n$ is the group of even permutations, we can choose $B = (1, 2, \ldots, n-2)$ and the ascending (resp. descending) projection permutes the entries of $x$ so that the first $n-2$ entries increase (resp. decrease) from left-to-right, and the last two entries are greater than or equal to all the other entries. If $x$ contains repeated entries then the last to entries can also be ordered to be increasing (resp. decreasing); otherwise their relative order depends on whether the permutation $s_{x'}$ which maps $i \mapsto x'_i$ for $1 \leqslant i \leqslant n$, is an even or odd permutation (see Section 2.3 for the definition of $x'_i$).

### C.3. The cyclic group

If $G = \mathbb{Z}_n \leqslant S_n$ is the cyclic group generated by the permutation $(1\ 2\ \cdots\ n)$, we can choose the base $B = (1)$. The ascending (resp. descending) projection cyclically permutes the entries of $x$ so that the first entry is less (resp. greater) than or equal to all other entries of $x$.

### C.4. The dihedral group

If $G = D_n \leqslant S_n$ is the dihedral group generated by

$$s_1 = (1\ 2\ \cdots\ n) \quad \text{and} \quad s_2 = (2\ n)(3\ (n-1))(4\ (n-2)) \cdots,$$

we can choose base $B = (1,2)$. The ascending (resp. descending) projection cyclically permutes the entries of $x$ via $s_1$ so that the first entry is less (resp. greater) than or equal to all other entries of $x$, and then if the final entry is less (resp. greater) than the second entry, it applies the permutation $s_2$.

### C.5. Products of groups acting on products of spaces

Suppose $G = \prod_{j=1}^{r} H_j$ where $H_j \leqslant S_{n_j}$ acts on $\bigoplus_{j=1}^{r} \mathbb{R}^{n_j}$ by each $H_j$ acting by permutations on the corresponding space $\mathbb{R}^{n_j}$ and trivially everywhere else. Let

$$B_j = \left(b_j^{(1)}, \ldots, b_j^{(k_j)}\right) \subset \{1, \ldots, n_j\} = N_j$$

be a base for $H_j$ acting on $\mathbb{R}^{n_j}$, then

$$B = \left(b_1^{(1)}, \ldots, b_1^{(k_1)}, b_2^{(1)}, \ldots, b_2^{(k_2)}, \ldots, b_r^{(1)}, \ldots, b_r^{(k_r)}\right)$$

is a base for $G$. Let $\pi_{j\uparrow} \colon \mathbb{R}^{n_j} \to \mathbb{R}^{n_j}$ be the ascending projection corresponding to $B_j$. Then define $\pi_\uparrow = \bigoplus_{j=1}^{r} \pi_{j\uparrow}$, to be the projection which equals $\pi_{j\uparrow}$ when restricted to $\mathbb{R}^{n_j}$. Similarly $\pi_\downarrow = \bigoplus_{j=1}^{r} \pi_{j\downarrow}$.

### C.6. Products of groups acting on tensors of spaces

Suppose $G = \prod_{j=1}^{r} H_j$ where $H_j \leqslant S_{n_j}$ acts on $\bigotimes_{j=1}^{r} \mathbb{R}^{n_j}$ by each $H_j$ acting by permutations on the $j$th component of $\bigotimes_{j=1}^{r} \mathbb{R}^{n_j}$, and trivially on the other components. For each $1 \leqslant j \leqslant r$ let $B_j = \left(b_j^{(1)}, \ldots, b_j^{(k_j)}\right) \subset \{1, \ldots, n_j\} = N_j$ be a base for $H_j$ acting on $\mathbb{R}^{n_j}$, and furthermore (for convenience) assume that $b_j^{(1)} = 1$. Then choose $B \subset \prod_{j=1}^{r} N_j =: N$ to be

$$B = \left((1, \ldots, 1),\ \left(b_1^{(2)}, 1, \ldots, 1\right), \ldots, \left(b_1^{(k_1)}, 1, \ldots, 1\right),\right.$$

$$\vdots \qquad\qquad \vdots$$

$$\left.\left(1, \ldots, 1, b_r^{(2)}\right), \ldots, \left(1, \ldots, 1, b_r^{(k_r)}\right)\right),$$

where a 1 in the $j$th position of an element of $B$ should be thought of as $b_j^{(1)}$. Suppose $x = (x_{l_1 \cdots l_r})_{l_1 \cdots l_r} \in \bigotimes_{j=1}^{r} \mathbb{R}^{n_j}$, and let $x'$ be as in Section 2.3. Choose $(m_1, \ldots, m_r) \in N$ to

be the index in the $G$-orbit of $\mathbb{1} = (1, \ldots, 1)$ with minimal entry in $x'$. For $1 \leqslant j \leqslant r$ define $x'_j \coloneqq (x'_{m_1 \cdots l_j \cdots m_r})_{1 \leqslant l_j \leqslant n_j} \in \mathbb{R}^{n_j}$, which is the restriction of $x'$ to the $\mathbb{R}^{n_j}$-vector containing the entry $x'_{m_1 \cdots m_j \cdots m_r}$. Then define

$$\phi_\uparrow \colon \bigotimes_{j=1}^r \mathbb{R}^{n_j} \to G \colon x \mapsto (\phi_{1\uparrow}(\hat{x}_1), \ldots, \phi_{r\uparrow}(\hat{x}_r)),$$

where $\phi_{j\uparrow} \colon \mathbb{R}^{n_j} \to H_j$ is the function defined for $H_j$ acting on $\mathbb{R}^{n_j}$, and similarly define $\phi_\downarrow(x)$. Then as before, $\pi_\uparrow(x) \coloneqq \phi_\uparrow(x) \cdot x$ and $\pi_\downarrow(x) \coloneqq \phi_\downarrow(x) \cdot x$.

## Appendix D. Proof of Theorem 6

The idea of the proof is as follows. In Appendix D.1 we shall outline an equivalence between subgroups of $S_n$ acting on $\mathbb{R}^n$ by permuting coordinates, and them acting on $S_n$ by multiplication. This will provide a dictionary between certain combinatorially defined fundamental domains and sets of coset representatives satisfying simple algebraic properties, Theorem 17. We will then outline the work from (12) in Appendix D.3 which gives an algorithm to find a set of coset representatives for an arbitrary subgroup of $S_n$. The main work is then to show this algorithm, with modifications, can produce a set of coset representatives with the desired algebraic properties so that it corresponds to a fundamental domain, culminating in Theorem 24. Finally we will show in Theorem 25 that the algorithm outlined in Section 2.3 indeed produces a projection onto this fundamental domain.

### D.1. Actions on $\mathbb{R}^n$ and $S_n$

Recall we have the group $S_n$ acting on $\mathbb{R}^n$ on the left by $s \cdot (x_i)_i = \left(x_{s^{-1}(i)}\right)_i$. We also have the normal action of $S_n$ on itself on the left by group multiplication: $s$ acts on $t$ by $s \cdot t = st$ for any $s, t \in S_n$. Here we shall show that in some sense these actions are equivalent. This correspondence is known, at least to experts, so we will only outline the essential points.

Let $x \in \mathbb{R}^n$ be a point, all of whose entries are distinct, and notice the set of such points is open and dense in $\mathbb{R}^n$. Define a function which changes the $i$th entry $x_i$ of $x$ to the integer $|\{1 \leqslant j \leqslant n \mid x_j \leqslant x_i\}|$. The result will be a list of the integers $1, \ldots, n$ in the same relative order as the entries of $x$, and we denote the set of all such points $C$. We can think of $C$ as a discrete subset of $\mathbb{R}^n$, and the left action of $S_n$ on $\mathbb{R}^n$ restricts to a left action on $C$. Notice also that this map $\mathbb{R}^n_{\mathrm{dist}} \coloneqq \{x \in \mathbb{R}^n \mid \text{all entries are distinct}\} \to C$ is continuous. In other words the set of connected components of $\mathbb{R}^n_{\mathrm{dist}}$ is in one-to-one correspondence with $C$, and indeed each component contains a point in $C$, its *representative point*. We call these connected components *chambers*, and given $c \in C$ we will write $[c] \subset \mathbb{R}^n$ for the corresponding chamber. The following is easy to check.

**Lemma 13** *Each chamber is a fundamental domain for the action of $S_n$ on $\mathbb{R}^n$.*

The action of $S_n$ on $\mathbb{R}^n$ preserves an $(n-1)$-simplex in the orthogonal complement of the vector $(1, \ldots, 1)$. In Figure 3 we show the 3-simplex preserved by $S_4$, and use it to visualise the $24 = |S_4|$ chambers in this case.

On the other hand, we can view each element of $C$ as a permutation in $S_n$ written in *in-line notation*. This means if $c = (c_i)_i$, as a permutation it sends $i$ to $c_i$ for each $i \in \{1, \ldots, n\}$. Thus $S_n$ is in one-to-one correspondence with $C$. In fact, it is better in our situation to modify

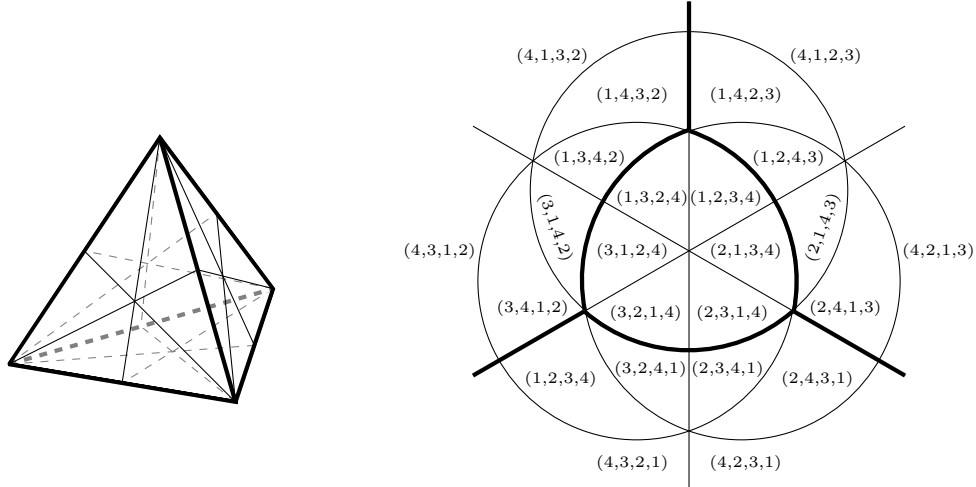

Figure 3: On the left is boundary of a 3-simplex, each small triangle corresponds to the intersection of this with a chamber. On the right the picture has been stereographically projected to the plane for the purposes of illustration, and each chamber is labelled by the representative element of $C$.

this correspondence by inverting elements of $S_n$ via the map $\rho\colon S_n \to C\colon s \mapsto (s^{-1}(i))_i$. The equivalence of the left action of $S_n$ on $\mathbb{R}^n$ and the left action on itself comes in the following form. Let $s, t \in S_n$, and consider the action of $s$ on $\rho(t)$:

$$s \cdot \rho(t) = s \cdot (t^{-1}(i))_i = (t^{-1}(s^{-1}(i)))_i = ((st)^{-1}(i))_i = \rho(st) = \rho(s \cdot t).$$

Given any subgroup $G \leqslant S_n$, the map $\rho$ defines an equivalence between $G$ acting on $\mathbb{R}^n$, which restricts to an action of $G$ on $C$, and $G$ acting on $S_n$ by left multiplication. We can use this equivalence to convert a set of right coset representatives for $G$ in $S_n$ into a complete set of orbit representatives (defined in Section 2.1) for $G$ acting on $\mathbb{R}^n$.

**Proposition 14**  *Let $R$ be a set of right coset representatives for $G \leqslant S_n$, then the set $\overline{\mathcal{F}} = \bigcup_{r \in R} \overline{[\rho(r)]}$ is a complete set of orbit representatives for $G$ acting on $\mathbb{R}^n$, where $\overline{[\rho(r)]}$ is the closure of the chamber containing $\rho(r)$.*

**Proof**  Since $\mathbb{R}^n_{\mathrm{dist}}$ is dense in $\mathbb{R}^n$ and $G$ acts by continuous maps which leave $\mathbb{R}^n_{\mathrm{dist}}$ invariant as a set, it suffices to show that $\bigcup_{r \in R} [\rho(r)]$ is a complete set of orbit representatives for $G$ acting on $\mathbb{R}^n_{\mathrm{dist}}$. In fact $G$ simply permutes the components of $\mathbb{R}^n_{\mathrm{dist}}$ so it suffices to show that $\bigcup_{r \in R} \rho(r)$ is a complete set of orbit representatives for the induced action of $G$ on $C$.

But now, $\rho$ is a bijection which exhibits an equivalence between the action of $G$ on $C$ and the action of $G$ on $S_n$ so we just need to show that $R$ is a complete set of orbit representatives for $G$ acting on $S_n$. The orbits of this action are precisely the right cosets of $G$, which completes the proof.  ∎

## D.2. Gallery connectedness and fundamental domains

Given a set of right coset representatives $R$ for $G \leqslant S_n$, the interior of $\overline{\mathcal{F}}$ as defined in the proposition will generally not be a fundamental domain because it will not be connected. We can reinterpret connectedness in terms of algebraic properties of $R$. First some geometric definitions.

**Definition 15** *Let $c, c' \in C$ be distinct, we say the chambers $[c]$ and $[c']$ are* adjacent *if $\overline{[c]} \cap \overline{[c']}$ has codimension 1. A* gallery *is a sequence of chambers $[c_1], \ldots, [c_k]$ such that consecutive chambers are adjacent. A set of chambers is called* gallery connected *if any two distinct chambers in the set can be connected by a gallery which is completely contained in the set. As a shorthand, we sometimes call a subset $C' \subset C$ gallery connected if the set $\{[c] \mid c \in C'\}$ is gallery connected.*

It turns out that the decomposition of $\mathbb{R}^n_{\text{dist}}$ into chambers corresponds to the chamber system of $S_n$ acting on its Coxeter complex, about which we will not elaborate here, but the interested reader should consult (4). The upshot of this viewpoint is two characterisations of adjacency of chambers.

**Lemma 16**  *Let $c \neq c' \in C$ and define $s = \rho^{-1}(c)$, $s' = \rho^{-1}(c')$. Then the following are equivalent:*

1. *The chambers $[c]$ and $[c']$ are adjacent*

2. *There is $1 \leqslant j \leqslant n - 1$ such that $s' = s(j \; j+1)$ where $(j \; j+1)$ is a transposition in $S_n$*

3. *The vectors $c$ and $c'$ differ by swapping exactly two entries which are consecutive integers*

**Proof** The equivalence of (1) and (2) is proved in, for example, Theorem I.5A of (4). To see the equivalence of (2) and (3), notice that $\rho(s(j \; j+1)) = ((s(j \; j+1))^{-1}(i))_i =: (c'_i)_i$. For $i \notin \{j, j+1\}$, $c'_i = s(i) = c_i$ (where $c := (c_i)_i$), whereas $c'_j = c_{j+1}$ and $c'_{j+1} = c_j$. ∎

The equivalence of (1) and 3 for the example of $S_4$ can be seen in (Figure 3). We will use this characterisation to prove Theorem 23 which is key to showing that the image of $\pi_\uparrow$ is connected. We are now in a position to upgrade Theorem 14 so that it produces a fundamental domain for the action of $G$. To state this, we define a *right transversal* of $G \leqslant S_n$ to be a minimal set of right coset representatives (ie a set containing exactly one element from every right coset).

**Proposition 17**  *Let $R \subset S_n$ be a right transversal for $G \leqslant S_n$ such that $\rho(R)$ is gallery connected. Then $\mathcal{F}$, the interior of $\bigcup_{r \in R} \overline{[\rho(r)]}$, is a fundamental domain for $G$ acting on $\mathbb{R}^n$.*

**Proof** By the definition, if $[c]$ and $[c']$ are adjacent, then the interior of $\overline{[c]} \cup \overline{[c']}$ will be connected. By induction on the length of galleries in $\{[\rho(r)] \mid r \in R\}$ it follows that $\mathcal{F}$ is connected. It is also open by definition.

By Theorem 14 we know that $\overline{\mathcal{F}}$ is a complete set of orbit representatives for $G$. Finally, suppose that some $G$-orbit meets $\mathcal{F}$ in at least two points, say $x$ and $x'$ and $g \in G$ is such that $g \cdot x = x'$. Since the $G$-action permutes the chambers, there are two possibilities:

1. There are coset representatives $r, r' \in R$ such that $x \in [\rho(r)]$ and $x' \in [\rho(r')]$.

2. There are coset representatives $r_1 \neq r_2, r_1' \neq r_2' \in R$ such that $x \in \overline{[\rho(r_1)]} \cap \overline{[\rho(r_2)]}$ and $x' \in \overline{[\rho(r_1')]} \cap \overline{[\rho(r_2')]}$.

In the first case we must have that $g \cdot [\rho(r)] = [\rho(r')]$, in which case it follows from Theorem 13 and the fact that $g \neq 1$, that $r \neq r'$. But then by the equivalence of the action with the action on $S_n$, we have that $g \cdot r = gr = r'$ and $r$ and $r'$ represent the same right coset of $G$. This contradicts the assumption that $R$ is minimal. In the second case we can similarly argue that $\{r_1, r_2\} \neq \{r_1', r_2'\}$ but $g \cdot \{r_1, r_2\} = \{r_1', r_2'\}$, again contradicting the minimality of $R$. In either case $g$ cannot exist. ∎

### D.3. An algorithm to find coset representatives

In this section we will summarise the main construction of (12) which gives an efficient algorithm to compute a right transversal for an arbitrary subgroup $G \leqslant S_n$. The first step, as it is to find a base $B \subset N$ for $G \leqslant S_n$. Set $B_0 = ()$, the empty tuple. We will assume that we have already constructed $B_{i-1}$ and computed $G_{i-1}$. If $G_{i-1} = \{1\}$, $B = B_{i-1}$ is a base and we are done. Otherwise, pick $b_i \in N$ with the largest orbit under $G_{i-1}$ and let $B_i$ be $B_{i-1}$ with $b_i$ appended.

Let $B = (b_1, \ldots, b_k)$ be a base and recall we define $G_0 = G$ and $G_i = G_{i-1} \cap \operatorname{Stab}_G(b_i)$ for $1 \leqslant i \leqslant k$. We also write $\Delta_i = b_i \cdot G_{i-1}$ for the orbit of $b_i$ under $G_{i-1}$. Recursively construct a partition $\Pi_i$ of $N$, starting with $\Pi_0 = \{N\}$. Denote by $\Gamma_i$ the element of $\Pi_{i-1}$ which contains $b_i$. One can check by induction that $\Gamma_i$ contains $\Delta_i$ as a subset. Define $\Pi_i$ by replacing $\Gamma_i$ in $\Pi_{i-1}$ by the non-empty subsets from the list: $\{b_i\}$, $\Delta_i - \{b_i\}$, and $\Gamma_i - \Delta_i$.

Now let $U_i$ be a right transversal for the group $\operatorname{Sym}(\Delta_i) \times \operatorname{Sym}(\Gamma_i - \Delta_i)$ in $\operatorname{Sym}(\Gamma_i)$, where $\operatorname{Sym}(\Omega)$ is the group of permutations of the set $\Omega$ (in the next section we will fix a particular choice for $U_i$), and finally let

$$H_i = \prod_{\Gamma \in \Pi_i} \operatorname{Sym}(\Gamma).$$

Then define $R = H_k U_k U_{k-1} \cdots U_1$, where for subsets $A, B \subset S_n$, $AB \coloneqq \{ab \mid a \in A, \ b \in B\}$.

**Theorem 18 ((12) §4)** *The set $R$ is a right transversal for $G \leqslant S_n$.*

### D.4. Gallery connected sets of coset representatives

We will now show how the method described above can be used to construct a right transversal $R$ for $G$ such that $\rho(R)$ is gallery connected. This is done by choosing a suitable base $B$, possibly re-indexing the set $N$, and choosing appropriate right transversals $U_i$ for $\operatorname{Sym}(\Delta_i) \times \operatorname{Sym}(\Gamma_i - \Delta_i)$ in $\operatorname{Sym}(\Gamma_i)$. We will prove Theorem 6 assuming $B$ and $N$ have been chosen in this way, and then in Appendix D.6 show that the assumptions on $B$ and $N$ can be dropped.

We described how to find a base for $G$ by appending more and more elements of $N$ to $B = ()$ until $G_k = \{1\}$ in Appendix D.3. The first assumption we make is that each new $b_i$ is minimal in the orbit $b_i \cdot G_{i-1}$ with respect to the normal ordering on $N$. We will call such

a base *orbit minimal*. We will use the following lemma to build gallery connected sets out of other gallery connected sets.

**Lemma 19** *Let $A_1, \ldots, A_l$ be subsets of $S_n$ so that each contains the identity permutation* (1), *and $\rho(A_i)$ is gallery connected for each $i$. Then $\rho(A_1 A_2 \cdots A_l)$ is gallery connected.*

**Proof** Notice that $A_1 A_2$ contains $(1)(1) = (1)$. Let $a \in A_1 A_2$, and choose $a_1 \in A_1$ and $a_2 \in A_2$ such that $a = a_1 a_2$. Since $\rho(A_1)$ and $\rho(A_2)$ are gallery connected, there are galleries $p_{a_1} \subset \rho(A_1)$ and $p_{a_2} \subset \rho(A_2)$ which connect $\rho((1))$ to $\rho(a_1)$ and $\rho((1))$ to $\rho(a_2)$ respectively. Then $a_1 \cdot p_{a_2}$ connects $\rho(a_1)$ to $\rho(a_1 a_2)$ in $\rho(a_1 A_2)$, and the concatenation $p_{a_1} * (a_1 \cdot p_{a_2})$ is a gallery which connects $\rho((1))$ to $\rho(a)$ in $\rho(A_1 A_2)$. Call this gallery $\tilde{p}_a$, its construction is illustrated in Figure 4. Now given $a, a'$ in $A_1, A_2$ the gallery $\tilde{p}_a^{-1} \cup \tilde{p}_{a'}$ (where $\tilde{p}_a^{-1}$ indicates $\tilde{p}_a$ traversed in reverse) connects $\rho(a)$ to $\rho(a')$ in $\rho(A_1 A_2)$, so $\rho(A_1 A_2)$ is gallery connected—the claim follows by induction on $l$. ∎

From the definition of $R$ in the previous section, if we show that $H_k$ and each of the $U_i$'s satisfy the hypotheses of this lemma, then it will follow that $R$ is gallery connected. We will first consider $H_k$. Notice that in fact, if $\Pi_k = \{N_1, \ldots, N_l\}$ then

$$H_k = \prod_{i=1}^{l} \mathrm{Sym}(N_i) = \mathrm{Sym}(N_1)\mathrm{Sym}(N_2)\cdots\mathrm{Sym}(N_l)$$

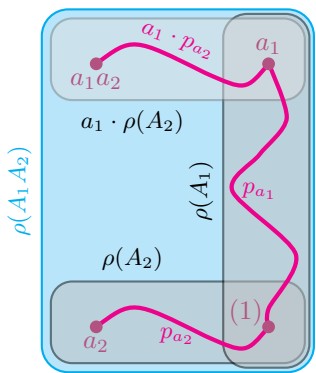

Figure 4: Building a gallery in $\rho(A_1 A_2)$.

can be written as a product of sets, again as in the lemma. Each $\mathrm{Sym}(N_i)$ contains (1), so we just need $\rho(\mathrm{Sym}(N_i))$ to be gallery connected for each $i$. It follows immediately from Theorem 16 that this is the case if and only if $N_i$ is a sequence of consecutive digits from $N$.

In general this will not be the case, however it can be readily achieved by re-indexing the set $N$. In fact we can do this so that each part of each partition $\Pi_i$ is a set of consecutive digits, which will subsequently aid in showing that $\rho(U_i)$ is gallery connected.

**Lemma 20** *We can re-index $N$ so that $b_i$ remains minimal in $\Delta_i$ and each part of $\Pi_i$ is a set of consecutive digits for $1 \leqslant i \leqslant k$.*

**Proof** We do induction on $i$: note that in $\Pi_0 = \{N\}$ the only part is a set of consecutive numbers. Assume that each element of $\Pi_{i-1}$ is a set of consecutive digits, in particular $\Gamma_i \in \Pi_{i-1}$ is a set of consecutive digits. Assume one of the three subsets $\{b_i\}$, $\Delta_i - \{b_i\}$ or $\Gamma_i - \Delta_i$ is non-empty and does not consist of consecutive digits, then by the minimality of $b_i$ both $\Delta_i - \{b_i\}$ and $\Gamma_i - \Delta_i$ must be non-empty and not consist of consecutive digits. Re-index the elements of $\Gamma_i$ so that overall the same set of digits is used, but now $b_i$ is the smallest, the next smallest digits lie in $\Delta_i - \{b_i\}$, and the remaining digits are in $\Gamma_i - \Delta_i$. ∎

### D.5. Choosing a right transversal $U_i$

Finally we want to choose the right transversals $U_i$ for $\mathrm{Sym}(\Delta_i) \times \mathrm{Sym}(\Gamma_i - \Delta_i)$ in $\mathrm{Sym}(\Gamma_i)$. Write $\Delta_i = \{d_1, d_2, \ldots, d_m\}$, and $\Gamma_i - \Delta_i = \{d_{m+1}, d_{m+2}, \ldots, d_{m+m'}\}$. For some integer $0 \leqslant l \leqslant \min\{m, m'\}$, choose $d_{j_1} < d_{j_2} < \cdots < d_{j_l}$ and $d_{m+j'_1} < d_{m+j'_2} < \cdots < d_{m+j'_l}$, and consider the product of transpositions

$$(d_{j_1}\ d_{m+j'_1})(d_{j_2}\ d_{m+j'_2}) \cdots (d_{j_l}\ d_{m+j'_l}) \in \mathrm{Sym}(\Gamma_i). \tag{4}$$

Define $\widetilde{U}_i$ to be the set of all such products for any choice of $l$, and indices $j_k$ and $j'_k$.

**Lemma 21** *((12) Lemma 2) $\widetilde{U}_i$ is a right transversal for $\mathrm{Sym}(\Delta_i) \times \mathrm{Sym}(\Gamma_i - \Delta_i)$ in $\mathrm{Sym}(\Gamma_i)$.*

Let $\tilde{u} \in \widetilde{U}_i$ have the form given in Equation (4). If $l = 0$, then $\tilde{u}$ is the identity and thinking of it as an element of $S_n \geqslant \mathrm{Sym}(\Gamma_i)$, we get $\rho((1)) = (1, \ldots, n)$. More generally $\rho(\tilde{u})$ will be the result of swapping each of the pairs $d_{j_k} \leftrightarrow d_{m+j'_k}$ in this vector, for $1 \leqslant k \leqslant l$. Let $g_{\tilde{u}} \in \mathrm{Sym}(\Delta_i) \times \mathrm{Sym}(\Gamma_i - \Delta_i)$ be the permutation such that $\rho(g_{\tilde{u}} \cdot \tilde{u})$ has its first $m$ entries in increasing order, and its last $m'$ entries in increasing order. Define $U_i = \{g_{\tilde{u}} \cdot \tilde{u} \mid \tilde{u} \in \widetilde{U}_i\}$.

**Lemma 22** *$U_i$ is a right transversal for $\mathrm{Sym}(\Delta_i) \times \mathrm{Sym}(\Gamma_i - \Delta_i)$ in $\mathrm{Sym}(\Gamma_i)$.*

**Proof** $\widetilde{U}_i$ contains exactly one element from each right coset of $\mathrm{Sym}(\Delta_i) \times \mathrm{Sym}(\Gamma_i - \Delta_i)$ in $\mathrm{Sym}(\Gamma_i)$. For $\tilde{u} \in \widetilde{U}_i$, the element $g_{\tilde{u}} \cdot \tilde{u} = g_{\tilde{u}}\tilde{u}$ lies in the same right coset as $\tilde{u}$ since $g_{\tilde{u}} \in \mathrm{Sym}(\Delta_i) \times \mathrm{Sym}(\Gamma_i - \Delta_i)$. Hence $U_i$ contains exactly one element from each right coset of $\mathrm{Sym}(\Delta_i) \times \mathrm{Sym}(\Gamma_i - \Delta_i)$ in $\mathrm{Sym}(\Gamma_i)$. $\blacksquare$

We want to show that $\rho(U_i)$ is gallery connected, and for that we will use the re-indexing of $N$ provided by Theorem 20. Recall that $b_i$ is the $i$th element of the base $B$; it follows from our construction of $U_i$ that

$$\rho(U_i) = \{(1, \ldots, b_i - 1, \overbrace{c_1, \ldots, c_m}^{\text{indexed by } \Delta_i}, \overbrace{c_{m+1}, \ldots, c_{m+m'}}^{\text{indexed by } \Gamma_i - \Delta_i}, b_i + m + m', \ldots, n) \mid$$
$$c_j \in \Gamma_i = \{b_i, b_i + 1, \ldots, b_i + m + m' - 1\} \text{ for all } 1 \leqslant j \leqslant m + m',$$
$$c_1 < \cdots < c_m, \text{ and } c_{m+1} < \cdots < c_{m+m'}\} \tag{5}$$

Since this is notationally rather cumbersome, we will abbreviate elements of $\rho(U_i)$ by

$$(c_1, \ldots, c_m \mid c_{m+1}, \ldots, c_{m+m'}),$$

where the first 'half' consists of entries indexed by $\Delta_i$, and the second 'half' consists of entries indexed by $\Gamma_i - \Delta_i$.

**Proposition 23** *Let $G \leqslant S_n$, $B$ be an orbit minimal base, and $N$ indexed so that each part of $\Pi_i$ is a set of consecutive digits. Then the set $\rho(U_i)$ is gallery connected.*

**Proof** To help simplify notation, we will not distinguish between points in $C$ and the chambers they represent. We will show that $\rho(U_i)$ is gallery connected by explicitly constructing a gallery which joins an arbitrary chamber

$$c = (c_1, \ldots, c_m \mid c_{m+1}, \ldots, c_{m+m'}) \in \rho(U_i)$$

to the chamber corresponding to the identity in $U_i$,

$$\hat{c} = \rho((1)) = (b_i, \ldots, b_i + m - 1 \mid b_i + m, \ldots, b_i + m + m' - 1).$$

Theorem 16 gives the condition for consecutive chambers in this gallery to be adjacent: they must differ by swapping two entries which are consecutive integers. Furthermore we will ensure this gallery remains in $\rho(U_i)$ throughout. This implies that after swapping the two entries, the two halves of $c$ must remain properly ordered. Taken together, this implies that:

> *The only swaps we can perform must switch the position of an entry in the left half with one in the right half, and these entries must be consecutive integers.*

Let $c \in \rho(U_i)$ be arbitrary, write $\hat{c}_j = b_i + j - 1$ for the $j$th entry of $\hat{c}$, and define

$$\delta(c) = \left( \sum_{j=1}^{m} c_j - \hat{c}_j \right) - \left( \sum_{j=m+1}^{m+m'} c_j - \hat{c}_j \right)$$

which measures the degree to which $c$ and $\hat{c}$ differ.

**Claim 1** $\delta(c) \geqslant 0$.
Let $j \leqslant m$, then since the entries in the left half of $c$ are ordered, distinct integers greater than or equal to $b_i$, $c_j \geqslant b_i + (j-1) = \hat{c}_j$ so each term in the first sum is non-negative. Similarly, for $j > m$ the entries in the right half of $c$ are ordered, distinct integers less than or equal to $b_i + m + m' - 1$, so $c_j \leqslant b_i + m + m' - 1 - (m + m' - j) = b_i + (j-1) = \hat{c}_j$ so each term in the second sum is non-positive. $\square$

As a remark, it follows from this claim that $\delta$ equals the $L^1$ distance between $c$ and $\hat{c}$. We shall perform a sequence of swaps as described above which have the effect decreasing the value of $\delta(c)$. Since $\delta(c) = 0$ implies that $c = \hat{c}$, the required gallery can be constructed by induction on $\delta(c)$. Assume $c \neq \hat{c}$, and let $j$ be the minimal index such that $c_j \neq \hat{c}_j$. Since the two halves of $c$ are ordered, $c_j$ is in the left half.

**Claim 2** $c_{j'} \coloneqq c_j - 1$ is in the right half of $c$.
Indeed suppose it is in the left half, then by the ordering on $c$, $j' < j$, and by the minimality of $j$, $c_{j'} = \hat{c}_{j'} = b_i + j' - 1$. But then $\hat{c}_j \neq c_j = c_{j'} + 1 = b_i + (j'+1) - 1 = \hat{c}_{j'+1}$, so $j \neq j' + 1$ since all entries of $\hat{c}$ are distinct. But now $c_{j'} < c_{j'+1} < c_j$ (by the ordering on $c$), which contradicts the fact that these entries are distinct integers, and $c_j - c_{j'} = 1$. $\square$

Thus, $c_j$ and $c_j - 1$ are entries in different halves of $c$ which are consecutive integers. Let $c'$ be the result of swapping these two entries in $c$, then

$$\delta(c) - \delta(c') = ((c_j - \hat{c}_j) - (c_{j'} - \hat{c}_{j'})) - ((c_{j'} - \hat{c}_j) - (c_j - \hat{c}_{j'})) = 2(c_j - c_{j'}) = 2 > 0$$

so performing the swap strictly decreases $\delta$. By induction, there is a gallery in $\rho(U_i)$ joining $c$ and $\hat{c}$, and hence $\rho(U_i)$ is gallery connected. $\blacksquare$

It follows directly from this proposition, Theorem 18, and Theorem 17 that $R$ as defined in Appendix D.3 corresponds to a fundamental domain.

**Corollary 24** *Let $G \leqslant S_n$, $B$ be an orbit minimal base, and $N$ indexed so that each part of $\Pi_i$ is a set of consecutive digits. Let $R$ be the right transversal for $G$ constructed above, then $\mathcal{F}$, the interior of $\bigcup_{r \in R} \overline{[\rho(r)]}$, is a fundamental domain for $G$ acting on $\mathbb{R}^n$.*

### D.6. Finishing off the proof

To complete the proof of Theorem 6 we have two tasks: first show that the map $\pi_\uparrow$ as defined in Section 2.3 has image in $\overline{\mathcal{F}} = \bigcup_{r \in R} \overline{[\rho(r)]}$, and so indeed projects onto a fundamental domain; and then remove the assumptions of orbit minimality and on how $N$ is indexed.

**Proposition 25** *Let $G \leqslant S_n$, $B$ be an orbit minimal base, and $N$ indexed so that each part of $\Pi_i$ is a set of consecutive digits. Then the image of $\pi_\uparrow$ lies in $\bigcup_{r \in R} \overline{[\rho(r)]}$.*

**Proof** It suffices to show that the image of $\mathbb{R}^n_{\text{dist}}$ lies in $\overline{\mathcal{F}} = \bigcup_{r \in R} \overline{[\rho(r)]}$. We claim that

$$\rho(R) = \{(c_j)_j \in C \mid \text{for } 1 \leqslant i \leqslant k, \ c_{b_i} \leqslant c_j \text{ for all } j \in \Delta_i\}.$$

It is clear that the definition of $\pi_\uparrow$ implies that the right hand side of this is the image of $\pi_\uparrow|_C$, so the proposition follows immediately from this claim.

Call the set on the right hand side $C'$, first we will show that $\rho(R) \subseteq C'$. By Equation (5) (note that the entries of $(c_j)_j$ there are indexed differently there) we can see

$$\rho(U_i) \subset \{(c_j)_j \in C \mid c_{b_i} \leqslant c_j \text{ for all } j \in \Delta_i\}.$$

Since $U_i \subset \text{Sym}(\Gamma_i)$, which fixes $b_{i-1}$ for $i \geqslant 2$, one can inductively check from the definition of $\rho$ that $\rho(U_k \cdots U_1) \subset C'$. Similarly, in the partition $\Pi_k$, each $b_i$ appears as a singleton, so $H_k$ also fixes $b_i$ for $1 \leqslant i \leqslant k$, hence $\rho(R) \subseteq C'$.

To establish the claim we just need to show that $|C'| = |\rho(R)|$; since they are finite sets, this implies that they are equal. On the one hand, since $\rho$ is a bijection, and using Lagrange's Theorem

$$|\rho(R)| = |R| = |\{\text{right cosets of } G \text{ in } S_n\}| = |S_n|/|G|.$$

On the other hand, each condition '$c_{b_i} \leqslant c_j$ for all $j \in \Delta_i$' decreases the size of $C$ by a factor of $|\Delta_i|$, so

$$|C'| = |C|/(|\Delta_1| \cdots |\Delta_k|).$$

Since $C$ is the bijective image of $S_n$ under $\rho$, $|C| = |S_n|$. By the Orbit-Stabiliser Theorem

$$|\Delta_i| = |b_i \cdot G_{i-1}| = |G_{i-1}|/|\text{Stab}_{G_{i-1}}(b_i)| = |G_{i-1}|/|G_i|,$$

Where the last equality follows from the definition $G_i = \text{Stab}_{G_{i-1}}(b_i)$. Therefore

$$|\Delta_1| \cdots |\Delta_k| = \frac{|G_0|}{|G_1|} \frac{|G_1|}{|G_2|} \cdots \frac{|G_{k-1}|}{|G_k|} = \frac{|G_0|}{|G_k|} = \frac{|G|}{|\{1\}|} = |G|$$

Hence $|C'| = |\rho(R)|$, which completes the proof. ∎

**Proof** [Proof of Theorem 6] Let $N = \{1, \ldots, n\}$, and choose $B$ a base for $G \leqslant S_n$, and $\varepsilon$ satisfying the conditions in Section 2.3. Let $s \in S_n$ be a permutation of $N$ such that $B^s := B \cdot s$ is an orbit minimal base, and each part of each partition $\Pi_i^s := \Pi_i \cdot s$ is a set of consecutive digits. That $s$ exists is clear by first permuting $k$ times so that $B$ is orbit minimal (note $b_i \notin \Delta_j$ for all $j > i$) and then applying Theorem 20. Write $b_i^s = b_i \cdot s$ so that $B^s = (b_1^s, \ldots, b_k^s)$.

Let $G^s = s^{-1}Gs$ be the conjugate of $G$ by $s$ in $S_n$, then for any $g \in G$ and $m \in N$,

$$(m \cdot s) \cdot g^s = ((g^s)^{-1}s^{-1})(m) = (s^{-1}g^{-1})(m) = (m \cdot g) \cdot s \tag{6}$$

where $g^s = s^{-1}gs$. In other words, permuting by $s$ and then acting by $G^s$ is the same as acting by $G$ and then permuting by $s$. It follows that $G_i^s := s^{-1}G_is = G_{i-1}^s \cap \mathrm{Stab}_{G^s}(b_i^s)$, and $\Delta_i^s := \Delta_i \cdot s = b_i^s \cdot G_{i-1}^s$.

Finally define $\phi_\uparrow^s$ and $\pi_\uparrow^s$ as in Section 2.3 with respect to $B^s$ and $\varepsilon$. We claim that for $x'$ as defined in Section 2.3, $\phi_\uparrow^s(x') = (\phi_\uparrow(x'))^s = g_{x'}^s$. Indeed by definition $\phi_\uparrow^s(x') = \tilde{g}_k \cdots \tilde{g}_1$ where $\tilde{g}_i \in G_i^s$ such that $\tilde{j} \cdot \tilde{g}_i = b_i^s$ and $\tilde{j} \in \Delta_i^s$ is chosen such that the $\tilde{j}$th entry of $(\tilde{g}_{i-1} \cdots \tilde{g}_1) \cdot x'$ is minimal among those entries indexed by $\Delta_i^s$. But now $\Delta_i^s = \Delta_i \cdot s$ means $\tilde{j} = j \cdot s$ (where $j \in \Delta_i$ is the index found in the definition of $\phi_\uparrow$). Thus

$$b_i^s = b_i \cdot s = (j \cdot g_i) \cdot s \overset{\text{Equation (6)}}{=} (j \cdot s) \cdot g_i^s = \tilde{j} \cdot g_i^s,$$

so we can certainly choose $\tilde{g}_i = g_i^s$. Then as claimed

$$\phi_\uparrow^s(x') = \tilde{g}_k \cdots \tilde{g}_1 = g_k^s \cdots g_1^s = (g_k \cdots g_1)^s = g_{x'}^s = (\phi_\uparrow(x'))^s.$$

Expanding out $\phi^s(x') = s^{-1}\phi(x')s$, we can now compute $\pi_\uparrow^s$ in terms of $\pi_\uparrow$ and $s$:

$$\pi_\uparrow^s(x) = \phi_\uparrow^s(x') \cdot x = s \cdot (\phi_\uparrow(x') \cdot (s^{-1} \cdot x)) = s \cdot \pi_\uparrow(s^{-1} \cdot x).$$

Writing $\mathcal{F}$ for the interior of the image of $\pi_\uparrow$, and $\mathcal{F}^s$ for the interior of the image of $\pi_\uparrow^s$, this implies $\mathcal{F}^s = s \cdot \mathcal{F}$ (because $s^{-1} \cdot \mathbb{R}^n = \mathbb{R}^n$). But Theorem 24 together with Theorem 25 says that $\mathcal{F}^s$ is a fundamental domain for $G^s$ and $\pi_\uparrow^s$ is a projection onto $\mathcal{F}^s$; so $\mathcal{F} = s^{-1} \cdot \mathcal{F}^s$ is a fundamental domain for $G$ and $\pi_\uparrow$ is a projection onto $\mathcal{F}$.

To prove the final claim of the theorem, that $\pi_\uparrow \colon \mathbb{R}^n \to \mathbb{R}^n$ is uniquely defined by the choice of $B$ and $\varepsilon$, we just need to show that a different choice of the elements $g_1, \ldots, g_k$ given $x \in \mathbb{R}^n$ does not change $\phi_\uparrow$. In fact $\phi_\uparrow$ is determined completely by what it does to the points $x' \in \mathbb{R}^n_{\text{dist}}$, and $\phi_\uparrow(x')$ will lie inside the fundamental domain (not on its boundary). By the definition of a fundamental domain, any different choice $g_1', \ldots, g_k'$ must necessarily combine to give the same element $g_{x'}$ (no non-trivial element of $G$ acts trivially), and hence $\phi_\uparrow$ is uniquely determined. ∎

## Appendix E. Other mathematical results

### E.1. Universal approximation theorem

The universal approximation theorem is a fundamental result in the theory of machine learning that any continuous function $\alpha \colon X \to \mathbb{R}^m$ on a compact subset $X \subset \mathbb{R}^n$ can be

arbitrarily well approximated by a neural network with one hidden layer. To state the theorem precisely, one needs to specify that closeness between two continuous functions $\alpha, \beta \colon X \to \mathbb{R}^m$ is measured by the $L^p$ norm (for $1 \leqslant p$) which is given by

$$\|f\|_p = \left( \int_X f(x)^p d\mu(x) \right)^{\frac{1}{p}},$$

where $\mu$ is the Lebesgue measure on $X \subset \mathbb{R}^n$. Then the distance between $\alpha, \beta$ is $\|\alpha - \beta\|_p$. Denote by $L^p(\mu, \mathbb{R}^m)$ the set of functions $f \colon X \to \mathbb{R}^m$ such that $\|f\|_p < \infty$, and by $\mathfrak{M}_{n,m}(\sigma)$ the set of functions $\mathbb{R}^n \to \mathbb{R}^m$ implemented by a neural network with activation function $\sigma \colon \mathbb{R} \to \mathbb{R}$, one hidden layer with arbitrarily many neurons and $m$ output neurons. The universal approximation theorem, as in Theorem 1 of (22), states that if $\sigma$ is unbounded and non-constant, then $\mathfrak{M}_{n,m}(\sigma)$ is dense in $L^p(\mu, \mathbb{R}^m)$ (for any compact subset $X$ of $\mathbb{R}^n$).

To obtain a $G$-invariant version of this theorem, assume that $X$ is a $G$-invariant subset of $\mathbb{R}^n$ and let $L^{pG}(\mu, \mathbb{R}^m)$ be the set of $G$-invariant functions $f$ on $X$ such that $\|f\|_p < \infty$. Denote by $\mathfrak{M}_{n,m}^G(\pi, \sigma)$ the set of functions of the form $\alpha \circ \pi$ where $\pi \colon X \to \mathbb{R}^n$ is a projection onto a fundamental domain, and $\alpha \in \mathfrak{M}_{l,m}(\sigma)$.

**Theorem 26 ($G$-invariant Universal Approximation Theorem)** *Let $X \subset \mathbb{R}^n$ be a compact $G$-invariant subset, $\sigma$ an unbounded and non-constant function and $\pi \colon \mathbb{R}^n \to \mathbb{R}^n$ a projection onto a fundamental domain. Then $\mathfrak{M}_{n,m}^G(\pi, \sigma)$ is dense in $L^{pG}(\mu, \mathbb{R}^m)$.*

**Proof** We need to show that given $\varepsilon > 0$, a $G$-invariant map $\alpha \colon X \to \mathbb{R}^m \in L^{pG}(\mu, \mathbb{R}^m)$, and a projection $\pi \colon \mathbb{R}^n \to \mathbb{R}^n$, there is a neural network $\overline{\beta} \in \mathfrak{M}_{l,m}(\sigma)$ such that $\|\alpha - \overline{\beta} \circ \pi\|_p < \varepsilon$. Let $\overline{\alpha} \colon \pi(X) \to \mathbb{R}^m$ be the restriction of $\alpha$ to the intersection of the closure of the fundamental domain with $X$, ie $\pi(X)$.

Since $\alpha \in L^{pG}(\mu, \mathbb{R}^m)$, it follows immediately that $\overline{\alpha} \in L^{pG}(\overline{\mu}, \mathbb{R}^m)$, where $\overline{\mu}$ is the restriction of $\mu$ to $\pi(X)$. By the universal approximation theorem (Theorem 1 in (22)) applied to $\pi(X)$, there is a network $\overline{\beta} \colon \mathbb{R}^n \to \mathbb{R}^m$ such that

$$\|\overline{\alpha} - \overline{\beta}\|_p < \varepsilon.$$

Furthermore, since $\alpha$ is $G$-invariant, $\alpha = \overline{\alpha} \circ \pi$, which implies that

$$\|\alpha - \overline{\beta} \circ \pi\|_p = \|\overline{\alpha} \circ \pi - \overline{\beta} \circ \pi\|_p = \|(\overline{\alpha} - \overline{\beta}) \circ \pi\|_p \leq \|\overline{\alpha} - \overline{\beta}\|_p < \varepsilon,$$

as required. ∎

## E.2. Computing the space of linear equivariant maps

A $G$-equivariant neural network (see for example (29; 9; 15)) consists of a series of $G$-equivariant linear maps $\lambda_i$ separated by some non-linear activation function $\sigma$, yielding $\beta = \lambda_k \circ \sigma \circ \cdots \circ \sigma \circ \lambda_1$. Restrictions are placed on the learnable parameters of each $\lambda_i$ to ensure they are $G$-equivariant. For example, if $\lambda_i \colon \mathbb{R}^n \to \mathbb{R}^n$ is equivariant with respect to $S_n$ acting on each copy of $\mathbb{R}^n$ by permuting coordinates, then it was shown in Lemma 3 of (41), that it must have the form

$$\lambda_i(x) = (a\mathbb{I} + b\mathbb{1}^T\mathbb{1})x, \tag{7}$$

where $a, b \in \mathbb{R}$ are learnable parameters, $\mathbb{I}$ is the identity matrix, and $\mathbb{1} = (1, 1, \ldots, 1)$. The main task is to describe the space of all $G$-equivariant linear maps $\lambda \colon \mathbb{R}^{n_1} \to \mathbb{R}^{n_2}$ which could map between layers in the neural network. Here we sketch an approach which is combinatorial and involves putting a $G$-invariant simplicial complex structure on $\mathbb{R}^{n_i}$ and applying the compatibility criterion Theorem 9 to the cells in the simplicial structure induced on the quotient space.

To slightly simplify matters for the purpose of exposition, assume that $G$ acts on $\mathbb{R}^{n_1}$ and $\mathbb{R}^{n_2}$ discretely and irreducibly by orthogonal matrices on each space. Because the action is orthogonal (ie it fixes the origin and preserves the Euclidean metric), it leaves the unit sphere $\mathbb{S}^{n_i-1}$ invariant in $\mathbb{R}^{n_i}$. The sphere is compact and the action is properly discontinuous, so it is possible to find some $G$-invariant triangulation of $\mathbb{S}^{n_i-1}$. This means a decomposition of $\mathbb{S}^{n_i-1}$ into $(n_i - 1)$-dimensional simplices (edges, triangles, tetrahedra, etc) such that each $g \in G$ acts on $\mathbb{S}^{n_i-1}$ by sending $k$-simplices to $k$-simplices for each $0 \leqslant k \leqslant n_i - 1$. Moreover, possibly after subdividing once, any two points in the interior of some $k$-simplex have the same stabiliser in $G$, and if some simplex $\sigma$ is a face of another simplex $\sigma'$, then $\mathrm{Stab}_G(\sigma') \leqslant \mathrm{Stab}_G(\sigma)$.

Now, the projection map $\pi_i \colon \mathbb{S}^{n_i-1} \to \mathbb{S}^{n_i-1}/G =: Q_i$ induces a simplicial structure on the quotient, and we can label each simplex $\overline{\sigma}$ in the quotient by the set of stabilisers of all simplices $\sigma \in \mathbb{S}^{n_i-1}$ which are mapped to $\overline{\sigma}$, $\pi_i(\sigma) = \overline{\sigma}$. In fact the label of $\overline{\sigma}$ defined in this way will be exactly the conjugacy class of $\mathrm{Stab}_G(\sigma)$ in $G$ for some (equivalently, any) $\sigma$ such that $\pi_i(\sigma) = \overline{\sigma}$. We can now try to construct a compatible map $\overline{\lambda'} \colon Q_1 \to Q_2$ by mapping simplices $\overline{\sigma}_1 \in Q_1$ to simplices $\overline{\sigma}_2 \in Q_2$ such that some element of the label $\overline{\sigma}_1$ is a subgroup of some element of the label of $\overline{\sigma}_2$ (note $\overline{\sigma}_1$ and $\overline{\sigma}_2$ need not have the same dimension) so that these maps glue together in a continuous way. This reduces the problem of checking the compatibility criterion on every point in $Q_1$ to only checking it on a finite number of simplices, and checking that the maps glue together, which is also a finite simplicial problem.

## Appendix F. Implementation

### F.1. Machine Learning Experiments

All neural networks were trained using Keras. SVMs were fitted using sci-kit learn. In each case the performance was averaged over 10 runs, and sample standard deviations are given in the tables in Section 3. The only exception to this is the training of SimpNet reported in Table 3, where we only trained a single run.

**Cayley tables** There are five isomorphism classes of groups with eight elements: $C_8$, $C_4 \times C_2$, $D_4$, $Q_8$, $C_2 \times C_2 \times C_2$. We generated a dataset of 40000 Cayley tables: a sixth of the tables were permutations of the group $C_8$, another sixth of the group $C_4 \times C_2$, another sixth of the group $D_4$, a quarter of the group $Q_8$, a quarter of the group $C_2 \times C_2 \times C_2$. The dataset was split into training and test dataset of equal size. Two models were compared: a fully connected neural network with two hidden layers of size 100 and 10 with activation function ReLU using the Adam optimiser with learning rate 0.001 and cross-entropy loss as a loss function training for 200 epochs; and a linear SVM.

**CICY** The neural network architectures and training parameters for the entries MLP, MLP + pre-processing, and Inception in Table 2 were taken from the given references. The

only difference we found is that in the case $\pi_{\mathrm{Dir}}$+Inception and $\pi_\uparrow$+Inception on the randomly permuted dataset, accuracy increased when *not* removing outliers from the training data, and we therefore decided to not remove outliers from the training data in this case, contrary to what was done in (14). The group invariant neural network had six equivariant layers (with four trainable parameters each) with 100 channels and cross-channel interactions, followed by sum pooling, and two fully connected layers with 64 and 32 neurons. No dropout was used. We experimented with max pooling, dropout, different numbers and sizes of layers and found the above parameters to work best. We found that test accuracy varied strongly for large batch sizes and eventually trained with batch size one. We randomly split the dataset into training and test sets of equal size.

**Classifying rotated handwritten digits**   Three neural networks were compared: first, a fully connected neural network with no hidden layers. Second, a fully connected neural network with two hidden layers of size 128 and 64 with activation function ReLU. Third, a small variation of the SimpNet architecture from (19), namely a convolutional neural network with the following layers: 13 convolutional layers with 64 channels in the first layer and 128 channels in all other layers, filter size $1 \times 1$ in layers 11 and 12 and filter size $3 \times 3$ in all other layers, max pooling layers with filter size $2 \times 2$ and strides $(2, 2)$ after the fourth and seventh layer and filter size $2 \times 2$ and strides $(1, 1)$ after the $9^{\mathrm{th}}$, $12^{\mathrm{th}}$, and $13^{\mathrm{th}}$ convolutional layers. All followed by one global max pooling layer, two hidden fully connected layers with 128 and 64 neurons respectively, and activation function ReLU after all hidden layers. We used the Adam optimiser with learning rate 0.001 and cross-entropy loss as a loss function. We trained for 100 epochs with early stopping if the training error does not decrease for 5 epochs.

### F.2. Computing combinatorial projections

In this section we will give algorithms to compute combinatorial projection maps for permutation group actions, and analyse the time and space complexity of these algorithms. These work for any permutation group $G$, although they are not the ones we used in Section 3 which employed more efficient *ad hoc* methods described in Appendix C. The general algorithms here fall into two parts: first are the algorithms which are applied as a one-off to compute data like a base and the orbits $\Delta_i$, and which run in $O(k^2 n^3)$ time, and $O(n^2 \log n)$ space, where $n$ is the dimension of the input space and $k$ is the size of the base, see subsection F.2.3. Second are the algorithms which actually implement the projection $\pi_\uparrow$, and which must therefore be run for each input datum. They do this in $O(k^2 n^2)$ time and $O(n^2 \log n)$ space. Since $\pi_\uparrow$ merely permutes the entries of a datum, it does not change the space required to store the input data.

Throughout we will maintain the same notation as before, where we are working with a subgroup of $S_n$ which acts by permuting the coordinates of $\mathbb{R}^n$ indexed by $N = \{1, \ldots, n\}$. As initial data we will assume we have a subgroup $G$ of $S_n$ given by a generating set of permutations. Moreover, we will assume that these permutations are given in *cycle* or *one-line* notation, so that each can be stored in $O(n \log n)$ space, and multiplying two permutations together can be performed in $O(n)$ time. Similarly, given $x \in \mathbb{R}^n$ and a permutation $g$, the point $g \cdot x$ can be computed in $O(n)$ time.

### F.2.1. COMPUTING INITIAL DATA

We will make use of the method of representing permutation groups introduced by Jerrum in (23) which we summarise. First we will explain the notations. Write $N = \{1, \cdots, n\}$, we will work with directed graphs (ie graphs whose edges have an orientation) of the form $(N, E)$, which have $N$ as their vertex set and $E$ as their edge set. These graphs will contain no edge loops, and at most one edge joining any pair of vertices, and so we can write $lm$ to denote an edge which starts at $l$ and ends at $m$. A *(directed) path* in $(N, E)$ is a sequence of vertices $l_0 l_1 \cdots l_m$ such that $l_j l_{j+1} \in E$ for each $j$, which is said to have *length* $m \geqslant 0$.

A directed graph is called a *branching* if it contains no paths of length $m \geqslant 1$ with the same start and end points and each vertex has at most one incoming edge. If $(N, E)$ is a directed graph, an *edge labelling* is a map $\sigma : E \to S_n : b_l b_m \mapsto \sigma_{lm}$ which assigns to each edge a permutation of $N$. This labelling extends to a labelling of paths by setting $\sigma_P = \sigma_{b_{l_0} b_{l_1}} \cdots \sigma_{l_{m-1} l_m} \in S_n$, where $P = l_0 l_1 \cdots l_m$. We can now give a Theorem-Definition of the representation of a permutation group, see Theorem 3.3 and Section 4 of (23).

**Theorem 27** *Let $G \leqslant S_n$ be a permutation group given by a set of generators, then there is an algorithm which yields a small base $B = (b_1, \ldots, b_k)$ for $G$ (see subsection F.2.3 for a quantitative discussion of what it means for a base to be small). Extend $B$ to a fixed ordering $\hat{B} = (b_1, \cdots, b_k, b_{k+1}, \ldots, b_n)$ on the set $N$. As before let $G_0 = G$, and define $G_i = \mathrm{Stab}_{G_{i-1}}(b_i)$ for $1 \leqslant i \leqslant k$. Then there exists an edge labelled directed graph $\Upsilon(G) = (N, E, \sigma)$ satisfying the following properties:*

*1. $\Upsilon$ is a branching*

*2. For all $b_l b_m \in E$*

    *(a) $l < m$ and $l \leqslant k$*

    *(b) $\sigma_{b_l b_m} \in G_{l-1}$*

    *(c) $b_l \cdot \sigma_{b_l b_m} = b_m$*

*3. The set $U_i := \{\sigma_P \mid P \text{ is a path in } \Upsilon \text{ starting at } b_i\}$ is a right transversal for $G_i$ in $G_{i-1}$ for each $1 \leqslant i \leqslant k$.*

*Then $\Upsilon(G)$ will be called a Jerrum representation of $G$. Given a generating set for $G$, a Jerrum representation can be computed alongside $B$ in $O(k^2 n^3)$ time and $O(n^2 \log n)$ space. This algorithm also computes the orbits $\Delta_i = b_i \cdot G_{i-1}$ for $1 \leqslant i \leqslant k$.*

**Remark 28** *The algorithm presented in (23) in fact assumes that $(1, 2, \cdots, n)$ has been chosen* a priori *to be the base, but it is straightforward to amend the algorithm to compute a more efficient base using the greedy algorithm mentioned at the start of Appendix D.3, compare with (3).*

### F.2.2. APPLYING $\pi_\uparrow$ TO INPUT DATA

Fix a permutation group $G$ and let $\Upsilon = \Upsilon(G)$ be a Jerrum representation for $G$. First we will prove a useful characterisation of the orbit $\Delta_{i+1}$.

**Lemma 29** *The orbit $\Delta_i$ is the set of $b_l \in N$ such that there exists a path $P$ in $\Upsilon$ which starts at $b_i$ and ends at $b_l$.*

**Proof** Note that by induction on $m$, for any path $P = b_{l_0} \cdots b_{l_m}$, (2) in Theorem 27 generalises to say

(a) $l_0 < l_m$ and $l_{m-1} \leqslant k$

(b) $\sigma_P \in G_{l_0-1}$

(c) $b_{l_0} \cdot \sigma_P = b_{l_m}$.

If $P$ starts at $b_i$, then $\sigma_P \in G_{i-1}$ and $b_i \cdot \sigma_P = b_l$ so $b_l \in \Delta_i$. Conversely, if $b_l \in \Delta_i$, there is some $g \in G_{i-1}$ such that $b_i \cdot g = b_l$. Note that the cosets of $G_i$ in $G_{i-1}$ are exactly the sets of the form $\{g \in G_{i-1} \mid b_i \cdot g = b\}$ for fixed $b \in \Delta_i$. Indeed $g, g' \in G_{i-1}$ are in the same coset iff $g'g^{-1} \in G_i$ iff $b_i \cdot (g'g^{-1}) = b_i$ iff $b_i \cdot g' = b_i \cdot g$. Since $U_i$ is a complete set of representatives, it contains an element from every coset, and hence an element which maps $b_i$ to $b_l$. Call this element $u_{b_l}$, then by definition there is some path $P$ which starts at $b_i$ such that $u_{b_l} = \sigma_P$, and by the observation above, the end point of $P$ must be $b_l$. ∎

With this Lemma we can give an algorithm to perform the main task in computing $\pi_\uparrow$, computing $\phi_\uparrow$ as a product of permutations $g_i \in G_{i-1}$, see Section 2.3.

**Proposition 30** *Given $x' \in \mathbb{R}^n$, all of whose entries are distinct, and a Jerrum representation $\Upsilon = \Upsilon(G)$ for $G$, there is an algorithm to compute $\phi_\uparrow(x')$ in $O(k^2 n^2)$ time and $O(n^2 \log n)$ space.*

**Data:** A point $x' \in \mathbb{R}^n$, a Jerrum representation $\Upsilon = \Upsilon(G)$, and the orbits $\Delta_i$.
**Result:** $\phi_\uparrow(x')$.

```
1 for 1 ≤ i ≤ k do                                    // Loop runs k times
2     Set j to be the index in Δ_i such that x'_j ≤ x'_l for all l ∈ Δ_i ;    // This will be the
      current working vertex in Υ
3     Set g_i = e;                              // This will accumulate edge labels from Υ
4     while j ≠ b_i do                                 // Loop runs at most |Δ_i| times
5         Set l to be the unique index in Δ_i such that lj is an edge of Υ
6         Set g_i = σ_{lj} g_i   Set j = l
7     end
8     Set x' = g_i · x'
9 end
10 Set φ_↑(x') = g_k ⋯ g_1
```
**Algorithm 1:** Computing $\phi_\uparrow$.

**Proof** The algorithm we will use is Algorithm 1. Recall the definition of $\phi_\uparrow(x')$. Assume $g_1, \ldots, g_{i-1}$ have already been found, $G_{i-1}$ acts transitively on $\Delta_i$, choose $j \in \Delta_i$ such that the $j$th entry of $(g_{i-1} \cdots g_1) \cdot x'$ is minimal among those entries indexed by $\Delta_i$. Choose $g_i \in G_{i-1}$ such that $j \cdot g_i = g_i^{-1}(j) = b_i$. Then define $\phi_\uparrow(x') := g_k \cdots g_1$.

The job of finding each $g_i$ is made easy by Theorem 29, since we just need to find a path $P$ in $\Upsilon$ joining $b_i$ to $j$, which is guaranteed to exist. Since $\Upsilon$ is branching, each vertex

has at most one incoming edge, and hence starting from $j$ and working backwards we are guaranteed to reach $b_i$. Making note of each edge label as we construct $P$, we can choose $g_i = \sigma_P$. This is achieved by the loop starting in Line 3.

We consider the time complexity of this algorithm. Following (23), $\Upsilon$ can be represented by an $n \times n$ array, whose $pq$-entry is NULL if $pq$ is not an edge of $\Upsilon$, and $\sigma_{pq}$ otherwise. Finding $l$ in Line 4 requires searching the $j$th column of this array for the unique non-NULL entry, whose index row index will be $l$, and so takes $O(n)$ steps. As mentioned above Line 5 also takes $O(n)$ steps, so the while loop at Line 3 takes $O(n|\Delta_i|)$ steps.

Finding $j$ in Line 1 requires $O(|\Delta_i|)$ steps, searching through each entry of $x'$ indexed by $\Delta_i$ and comparing it with the current minimal entry found; while Line 7 takes $O(n)$ steps. Thus the dominant step in the main for loop is the while loop. Overall then this for loop takes $O(kn \sum_{i=1}^{k} |\Delta_i|)$ steps, which is greater than the $O(kn)$ steps to compute Line 9. Noting that $|\Delta_i| \leqslant n$, this algorithm runs in $O(k^2 n^2)$ time.

As for space, $\Upsilon$ is an $n^2$ array, containing at most $n-1$ non-NULL entries (since the graph is has no cycles, its number of edges is bounded by $n-1$), each of which takes $O(n \log n)$ space to store. An efficient encoding can then use $O(n^2 \log n)$ space. The other significant space cost is storing the set of $g_i$'s, which takes $O(kn \log n)$ which is at most $O(n^2 \log n)$. ∎

**Theorem 31** *Given $x \in \mathbb{R}^n$, a perturbation vector $\varepsilon$, and a Jerrum representation $\Upsilon$ for $G$, there is an algorithm to compute $\pi_\uparrow(x)$ in $O(k^2 n^2)$ time and $O(n^2 \log n)$ space.*

**Data:** A point $x \in \mathbb{R}^n$, a perturbation vector $\varepsilon$, a Jerrum representation $\Upsilon = \Upsilon(G)$, and the orbits $\Delta_i$.
**Result:** $\pi_\uparrow(x)$.
**1** **if** $x \in \mathbb{R}\mathbb{1}$ **then**
**2** $\quad$ Set $d = 1$
**3** **else**
**4** $\quad$ Set $d = \min\{|x_i - x_j| \mid x_i \neq x_j\}$
**5** **end**
**6** Set $x' = x + d\varepsilon$
**7** Compute $\phi_\uparrow(x')$;                                    // Algorithm 1
**8** Set $\pi_\uparrow(x) = \phi_\uparrow(x') \cdot x$
$\quad$ **Algorithm 2:** Computing $\pi_\uparrow$.

**Proof** Algorithm 2 follows exactly the procedure outlined in Section 2.3. It is clear that Line 6 dominates in terms of both time and space complexity. ∎

### F.2.3. ANALYSING TIME COMPLEXITY

Naïvely one may assume that the size $k$ of the base $B$ is $O(n)$, indeed this is the case for $G = S_n$ or $A_n$ for example, but in practice we can do a lot better. Write $b(G)$ for the size of the smallest base for $G$, then in (3) Blaha showed that the greedy algorithm used in Theorem 27 to find a base will not necessarily yield a minimal base, but nearly—in particular $k$ will be $O(b(G) \log \log n)$.

When looking at permutation groups, it is natural to focus on the case of so-called *primitive* permutation groups, ie subgroups $G \leqslant S_n$ which act transitively on $N = \{1, \ldots, n\}$ such that there are no non-trivial $G$-invariant partitions. This is because arbitrary permutation groups can be built up out of primitive ones. In this setting Liebeck proved the following in (27).

**Theorem 32** *Let $G$ be primitive, and not $S_n$ or $A_n$, then there is some absolute constant $c$ such that $b(G) < c\sqrt{n}$.*

It follows that in this case, the base found above will have size $O(\sqrt{n} \log \log n)$. Combining this with the observation in Appendix C that for $S_n$ and $A_n$, $\pi_\uparrow$ can be computed using a sorting algorithm, we get the following.

**Theorem 33** *Let $G \leqslant S_n$ be primitive, then either*

- *$G = S_n$ or $A_n$: no initial data needs to be computed and $\pi_\uparrow$ can be computed in $O(n^2)$ time per datum (using worst case for quicksort); or*

- *Initial data can be computed in $O(n^4 (\log \log n)^2)$ time, and $\pi_\uparrow$ can be computed in $O(n^3 (\log \log n)^2)$ time per datum.*

### F.3. Computing Dirichlet projections

We focus on the case $G$ acts discretely on $\mathbb{R}^n$ by orthogonal matrices. Choose a point $x_0 \in \mathbb{R}^n$ which is only fixed by elements of $G$ which fix the whole of $\mathbb{R}^n$ point-wise. The map $\phi$ from Section 2.2 maps $x \in \mathbb{R}^n$ to the element in $G$ which minimises the Euclidean distance $d(g \cdot x, x_0)$. In this case, the inner product $\langle \cdot, \cdot \rangle$ is invariant. It is efficient to compute $\langle \cdot, \cdot \rangle$, which varies *inversely* with the Euclidean distance $d$ between points, so we can perform gradient descent to minimize

$$N(g \cdot x) := \langle g \cdot x, -x_0 \rangle = \frac{1}{2} \left( d(g \cdot x, x_0)^2 - |g \cdot x|^2 - |x_0|^2 \right).$$

Here the second equality comes from applying the cosine rule and the geometric definition of the inner product.

Our main application of the discrete gradient descent algorithm is for CICY matrices when $G = S_{12} \times S_{15}$. Since $|G| \approx 6 \times 10^{20}$ one cannot minimise a function over $G$ by simply evaluating it at all elements of $G$.

#### F.3.1. Approximations using gradient descent

It is natural to compute the minimiser of $N$ on a group orbit using gradient descent. The steps in the descent are restricted to the discrete $G$-orbit of the input point $x$, so we must define what a gradient is in this case. A *generating set* $T$ for $G$ is a subset such that any element $g \in G$ can be written as a finite product of elements in $T$, ie there are $t_1, \ldots, t_l \in T$ such that $g = t_1 \cdots t_l$. Two points $x, x' \in \mathbb{R}^n$ are *adjacent* with respect to $T$ if there is $t \in T$ such that $x' = t \cdot x$, so in particular, adjacent points are in the same $G$-orbit.

**Definition 34** *Given an action of a finite group $G$ on $\mathbb{R}^n$, a generating set $T$ of $G$, a function $N\colon \mathbb{R}^n \to \mathbb{R}$ and $x \in \mathbb{R}^n$, the* discrete gradient descent *is an approximation for*

$$\min_{g \in G} N(g \cdot x)$$

*and is defined iteratively as follows. Let $x_0 = x$. Given $x_i$, define*

$$x_{i+1} = \min_{t \in T \cup \{e\}} N(t \cdot x_i).$$

*The output of the algorithm is $x_i$ when $x_{i+1} = x_i$.*

Since we can get between any two points in $G$ by a finite sequence of steps by generators this algorithm always terminates. In general, there are many choices for a generating set $T$ resulting in different approximations for $\phi$. For $G = S_n$ a natural choice for a generating set is given by $T = \{(1\,2), (2\,3), \ldots, (n-1\,n)\}$. In particular, one has in this case $|T| = n - 1 \ll n! = |S_n|$. By taking the union of these generating sets for $S_n$ and $S_m$ one obtains a generating set of size $n + m - 2$ for $S_n \times S_m$.

Choosing a larger generating set increases the computational cost of the algorithm but potentially also its accuracy. For example, consider the set $T' = \{tt' \mid t, t' \in T \cup \{e\}\}$. This is a generating set for $S_n$ and again yields a generating set for $S_n \times S_m$ in a similar way. When applied to the CICY dataset, we found that choosing $T'$ instead of $T$ leads to a significant increase in computation cost, but not so in accuracy. Instead, we have used discrete gradient with different seeds.

For a $12 \times 15$ CICY matrix $x$, The seeds are $x_{km} := C_{12}{}^k x C_{15}{}^m$, where $C_i$ are cyclic permutation matrices $1 \le k \le 12$ and $1 \le m \le 15$. To each $x_{km}$, apply the discrete gradient descent algorithm above and pick the minimum of all seeds. The result is still a permutation of $x$ since $C_i$ are permutation matrices. This increases the computation cost by a constant factor $11 \times 14 + 1 = 155$ but has led to a significant accuracy boost.

We are unable to give a bound for the number of generators applied to an input until a local minimum is reached. Experiments on the CICY dataset show that this number is very low compared to the size of the group. On the original CICY dataset the average number of iterations is $\approx 17.4$, with standard deviation $\approx 15.6$ and maximum 163. On an augmented dataset, which contains 10 permutations of each matrix, the average number of iterations is $\approx 22.5$, with standard deviation $\approx 16.9$ and maximum 198.

### F.3.2. The existence of an exact algorithm

Computing a Dirichlet projection can be hard and we do not expect that in general there is a polynomial time algorithm to compute it. If such an algorithm exists, this would imply a solution to the graph isomorphism problem in polynomial time, and it is currently unknown whether such an algorithm exists. To see this, let $n \in \mathbb{N}$ and $\Gamma_1$, $\Gamma_2$ be graphs with adjacency matrices $A_1, A_2 \in \{0, 1\}^{n \times n}$. The symmetric group $S_n$ acts on the graphs by relabelling the vertices, which can be viewed as an action on $\{0, 1\}^{n \times n}$. Let

$$x_0 = \begin{pmatrix} 2^{n \cdot n - 1} & 2^{n \cdot n - 2} & \cdots & 2^{n \cdot n - n} \\ \vdots & \vdots & \ddots & \vdots \\ 2^{n-1} & 2^{n-2} & \cdots & 2^0 \end{pmatrix}$$

and denote the corresponding Dirichlet projection by $\pi : \{0,1\}^{n \times n} \to \{0,1\}^{n \times n}$. Because of our choice of $x_0$, for all $A \in \{0,1\}^{n \times n}$ there exists a unique $g \in S_n$ that minimises $\langle g(A), x_0 \rangle$. The element $g$ can be characterised as the permutation that makes the matrix $A$ as small as possible, when read as a binary number row by row.

The graphs $\Gamma_1$ and $\Gamma_2$ are isomorphic if and only if $A_1$ and $A_2$ are in the same orbit under the $S_n$ action, which is the case if and only if $\pi(A_1) = \pi(A_2)$. Therefore, if one is able to compute $\pi(A_i)$ for $i \in \{1,2\}$ in polynomial time, one can decide if $\Gamma_1$ and $\Gamma_2$ are isomorphic in polynomial time.

