# OpenReview forum: "Group invariant machine learning by fundamental domain projections"
_NeurIPS.cc/2022/Workshop/NeurReps — NeurReps 2022 Poster_

### Official Review · Reviewer_WtUF · 2022-10-15
**Invariant machine learning over the quotient space: useful contribution**

**Confidence:** 4
**Soundness:** 4
**Presentation:** 4
**Contribution:** 4
**Overall Rating:** 8

**Summary:**

This paper defines an efficient approach for group invariant and equivariant machine learning with respect to permutation subgroups. They give an efficient algorithm for mapping neural network inputs onto the orbit space of the input space X with respect to the underlying permutation symmetry G. This approach is evaluated on a Hodge number prediction problem, as well as Cayley table prediction and rotated MNIST, and outperforms alternatives.

**Questions:**

1. What is the precise effect of adding random noise to avoid repeated elements in the input vector? Is this a method solely to break symmetries (that is then undone after an orbit representative is computed)?

2. In what sense is this method formally a “projection”, particularly in light of the previous question? (Given the fundamental domain provided by the algorithm, is there a distance metric under which this mapping computes the closest point in the fundamental domain?)

3. Is there an explicit relation to frame averaging? In particular, can the orbit projection be expressed as summing over a particular equivariant frame? It would be helpful for the reader to make this explicit in the paper.

4. Although I believe the randomized orbit separation method of Dym and Gortler 2022 was not worked out for arbitrary permutation subgroups, can the authors clarify on the advantages and disadvantages of this method compared to theirs?

5. Other than computability, what is the advantage over polynomial invariants? Remark 12 in the appendix says that “they significantly distort the training data, leading to low accuracy. To avoid this one must find an isometric embedding…this is even more difficult, and likewise significantly increases the ambient dimension of the training data”. Can the authors cite or demonstrate this assertion? What is an explicit comparison between the runtime of the proposed algorithm and the runtime of computing polynomial invariants?

6. How substantial were the adaptions made to the coset representative algorithm of Dixon and Mated 1988 (11)? The authors could expand on this comment on page 5, to better highlight their contributions.

7. How does the proposed algorithm circumvent the graph isomorphism problem, when the Dirichlet algorithm (as shown in the appendix) does not? More broadly, what pre-processing procedure would this paper imply for learning on graphs (with second order permutational symmetry), and how does it perform relative to existing graph neural network (GNN) architectures?

**Limitations:**

The experiments are a bit niche (Hodge number prediction) for a broader machine learning community. Moreover, the baselines for comparison should ideally include polynomial invariants, or a remark on why computing these invariants is truly unfeasible otherwise. In fact, it would be very helpful to compare this paper’s method, deep sets, polynomial invariants, and Dym and Gortler’s invariant embeddings on symmetric group invariant learning tasks, to validate the claim that mapping inputs to a fundamental domain (which is connected) is easier for learning than other methods for separating orbits. I would also like to better understand the effect of adding random noise to break self-symmetric objects; will this result in a degradation in performance on e.g. a dataset of only self-symmetric inputs?

**Recommended Decision:**

3: Accept

**Relevance:**

4: Highly relevant

**Strengths And Weaknesses:**

Originality:
The work answers a fundamental and useful question in equivariance (what is the “right” way of computing orbit representatives with respect to a permutation subgroup?), drawing on and adapting an algorithm for computing coset representatives from 1988. It is related to recent work on computing low-dimensional invariant embeddings (Dym and Gortler 2022), and also seems like a more general case (for general permutation subgroups) of frame averaging (puny et al 2022) — although its contributions are novel relative to these works, it would be extremely helpful to contextualize and compare them.

Quality:
The submission seems to be technically sound. The experimental claims are supported by their experiments.

Clarity:
The submission is exceptionally clearly-written and well-organized. (As a minor suggestion, given the length of the appendix, it would be helpful if it had its own outline or table of contents.)


Significance:
This work is a valuable contribution to the equivariant learning community, which has many different paradigms for learning permutation equivariant functions, but did not previously have a scalable method for enforcing equivariance under arbitrary permutation subgroups. It also connects naturally with previous work on frames and orbit separators. The experiments (particularly the Hodge number prediction task) may not be especially compelling for a broader machine learning audience, but suffice as a powerful proof of concept. Although the authors highlight the extension of this work to Lie groups as a future direction, I find their contribution on permutation groups to be more interesting and less obvious, since the orbit space under a Lie group action should have a clear manifold structure.


**Submission Track:**

Proceedings Paper (9 Page)

---

> ### Author Response · Authors · 2022-10-31
> **Our approach vs other G-invariant architectures**
>
> Questions:
>
> We reply to the questions in the order they were asked
> (1) The reviewer is correct, the noise is only there to temporarily break the symmetry so that the projection can be uniquely defined. As it is effectively undone in the final definition of \pi, the noise has no effect on the actual data which eventually gets fed into the machine learning model.
> (2) It is a projection in the sense that pi^2 = pi. The Dirichlet fundamental domain is exactly defined so that the closest point condition is satisfied.
> (3) We compare this with the article “Frame averaging for invariant and equivariant network design” now. In the ideal situation that the frame F takes a single group element as its value everywhere, this defines the fundamental domain projection mapping X to F(X)X. This ideal situation almost never occurs and then our approaches are different. In our main example we did not find a frame that is everywhere smaller than the whole group, which makes the “Frame” approach intractable in this case.
> (4) Thank you for bringing this paper to our attention, we were not aware of it. Similar ideas are involved and we need extra time to study this. We briefly mention some differences that we spotted after a first reading. Advantages of our approach: works for many groups, projections have been worked out for many discrete groups, including the case of row/column permutations on matrices (unlike theirs). Advantages of their approach: works for continuous groups, their separating maps are continuous, unlike ours. We have also added a sentence to the introduction.
> (5) The embedding dimension for polynomial invariants grows rapidly, making it infeasible for most practical purposes. For example, in the case of Z_4 acting on images with 4xk pixels, we computed the embedding dimension with the Sage function ‘G.invariant_generators()’ (from the finitely generated matrix groups class)
> for k=1,...9 to grow like 7, 31, 84, 180, 335, 567, 896, 1344, 1935. Extrapolating, this gives a target dimension for the 28x28 pixel images (when k=196) of ~1.34x10^8 - it is computationally infeasible to compute this many polynomial invariants.
>
> Even provided one can compute the polynomial invariants, the embedding significantly distorts the geometry of the feature space. In informal experiments we found that this led to very bad accuracies in the case of Z_4 rotating 4 pixel images (i.e. when there are 7 polynomial invariants). This distortion can be quantified by integrating the difference between the induced metric on the image of the feature space and the pushforward of the flat metric on the feature space. Comparing the polynomial invariants map to a similar projection which is designed to try to preserve the geometry of the feature space as much as possible, the former has 30 times the distortion of the latter. The details are soon to appear in the PhD thesis of the third author.
>
> Some details on this have been added to the paper in Remark 11.
> (6) We have added some clarifications after Example 3.
> (7) The Dirichlet projection algorithm does solve the graph isomorphism problem, but it does so in exponential time. Our approach would amount to taking the adjacency matrix of some graphs as input, projecting them onto a fundamental domain, and then training an ML model to these projections. There is no obvious ML model to take here, for example one would need to require all graphs to have the same number of vertices to apply a neural network. In these contrived settings we might be able to carefully construct datasets on which GNNs fail and our approach is successful, but generally we see no way to compare the two approaches and concede that GNNs are far more versatile when it comes to graph problems.

---

### Official Review · Reviewer_8S41 · 2022-10-15
**Interesting ideas for some symmetric group actions, and very clear exposition. not sure about how widely applicable this is.**

**Confidence:** 4
**Soundness:** 3
**Presentation:** 4
**Contribution:** 3
**Overall Rating:** 6

**Summary:**

The author(s) consider finding orbit representatives for certain types of group actions and apply this to produce group-invariant or equivariant models. Given an element of an input set $X$, and a group $G$ which acts on $X$, the authors aim to find a representative for each coset in $X/G$. They explain how to do this for a few very simple cases first, such as when $G = S_n$ and $X = \mathbb{R}^n$, and $G$ acts by coordinate permutations. As $\mathbb{R}$ is totally ordered, a reasonable choice of coset representatives can be found by sorting the coordinates of an input in ascending order. This is just a very simple example and perhaps the more interesting contribution is pointing out how a canonical choice of coset representative can be found when $G$ is an interesting subgroup of $S_n$. They mention an algorithm and claim that it is efficient in the size of the group $|G|$, though the details are mostly deferred to an appendix. They apply this to some examples, one trivial case of classifying groups of order 8 from their cayley table, and some more interesting cases of discretely-rotated MNIST digits and some applications in string theory.

**Questions:**

- What is the "closure" of $\mathcal{F}$ ? This term is used on page 2 and this far in the paper, $X$ is just a set, so I'm not sure if there's a topology assumed or what.

- What is the exact runtime / complexity of the algorithm, given the size of a group G? what about when G is way too large, and we can just write down the generators? is there a way to avoid scaling with the size of G and just the number of generators instead?

- Can you explain more about how this generalizes to continuous and / or infinite groups?

**Limitations:**

- A huge amount of content is deferred to the appendix. I think some of this information could be summarized in the main text without making it much longer: for example the precise dependence of the worst-case runtime of their algorithm for the S_n subgroups should be mentioned in the main text even if the details of the algorithm have to wait until the appendix.

- I think the experiment / example with the groups of order 8 is a bit misleading. If I have understood what they are doing here, a hash table / look up table would do in this case, and there is no held-out test set (there are only 5 points (groups) in the first place). I.e. you will always see the exact same matrix for the cayley table / multiplication table after applying the fundamental domain technique. This is not necessarily a weakness of mapping to the fundamental domain but it's a bit misleading to show an accuracy like $96\%$ when computing a checksum would get you 100%.

It would be far more interesting to consider larger groups than 8, and more nontrivial properties such as commutativity, with properly held-out test set. However I am concerned that the technique would not work well in this case.


**Recommended Decision:**

2: Borderline

**Relevance:**

4: Highly relevant

**Strengths And Weaknesses:**

- The exposition is very clear and easy to follow. The case of coordinate permutations is indeed an interesting and widely-applicable case.

- It's somewhat difficult to ascertain the specific contribution that the authors claim to be making. In 1.2 they seem to suggest it is their general approach of mapping to the fundamental domain. However they describe how to do this only for specific finite group actions pf the symmetric group.
My best guess at what the contribution of this paper is is that they are introducing this concept and terminology to the ML community and showing a few techniques, which are perhaps well-known in the maths community, for implementing it in very limited cases. As a secondary contribution the string theory applications are interesting and it would be good to provide more details of that dataset.

- (Nitpick) I would prefer the authors use a more verbose citation style. I am unsure whether the citation style they use of just listing numbers is permitted by the Neurreps style guide. In any case, I'm used to seeing name / dates and it's a bit easier to keep track that way. Perhaps I am just old-fashioned.

- The claims at the beginning seem overly-broad, when in fact the paper only ends up tackling the specific case of finite group actions, specifically subgroups of the symmetric group acting by coordinate permutations. I think they would be better off being more clear about their limitations up front. If you read the beginning it sounds more like they will give a general algorithm for any group action.

**Submission Track:**

Proceedings Paper (9 Page)

---

> ### Author Response · Authors · 2022-10-31
> **Topological assumptions and infinite group actions**
>
> Strengths and weaknesses:
> Reviewer:  ‘As a secondary contribution the string theory applications are interesting and it would be good to provide more details of that dataset.’
> Reply:  We have added more details on Hodge numbers and CICYs to the beginning of 3.2.
>
> Reviewer: ‘I am unsure whether the citation style they use of just listing numbers is permitted by the Neurreps style guide.’
> Reply: It looks like the new citation style in the template is numbers rather than authoryear, so no change needs to be made.
>
> Reviewer: ‘I think they would be better off being more clear about their limitations up front. If you read the beginning it sounds more like they will give a general algorithm for any group action.’
> Reply:  We have added the following sentence to the introduction: ‘’We provide a general framework to define this preprocessing step for general group actions and a concrete implementation for finite groups acting by coordinate permutations.’
>
>
> Questions
> We reply to questions in the order they were asked
>
> (1) The setting of the paper is of a discrete group acting by isometries on a Riemannian manifold, or sometimes just on R^n. In principle, the notion of a fundamental domain makes sense in a much broader setting: something like a properly discontinuous action of a group by homeomorphisms, but one must always assume some kind of topology has been fixed for the definition of a fundamental domain to make sense. Without a topology, one can only talk of “a complete set of unique orbit representatives”, which then becomes a fundamental domain if you impose that the set has the discrete topology.
> (2) We have added a summary of time complexity of the algorithm to the paragraph before 2.4.
> (3) In the infinite case, we are lacking an analogue of Theorem 4, i.e. a general theorem for constructing a parametrisation of the quotient space, but there are constructions for two cases: Discrete infinite groups acting properly discontinuously and Lie groups of positive dimension acting smoothly. In the first case Dirichlet fundamental domains still exist and the algorithm for approximate Dirichlet projection terminates in finite time. For some specific examples of infinite groups, for example the wallpaper groups discussed by Cohen and Welling in “Group Equivariant Convolutional Networks”, it is relatively straightforward to see how one might define an analogue of the combinatorial projection. First one would project onto a fundamental parallelogram for the translation subgroup, and from there onto a fundamental domain for the whole group. This last step becomes a finite problem, possible with sufficient understanding of the group, since the translation subgroup has finite index. The analogue of a fundamental domain for Lie group actions is the notion of a slice. Slices have the great advantage of having a lower dimension than the input space itself, cf. Sec. 2.13 of [Berndt J, Console S, Olmos CE, Submanifolds and holonomy. CRC Press; 2016]. Locally, one can construct slices as the normal space of a group orbit, which is a global construction in many concrete cases (where the injectivity radius equals the diameter of the manifold).
>
>
> Limitations:
> Reviewer:‘[...] the worst-case runtime of their algorithm for the S_n subgroups should be mentioned in the main text even if the details of the algorithm have to wait until the appendix.’
> Reply: Done, see point 2 under questions above.
>
> Reviewer: ’I think the experiment / example with the groups of order 8 is a bit misleading. [...]’
> Reply: The case of groups of size 8 is very much a toy example, and we chose to stick with it rather than choosing a more complicated version since it has been studied previously in the literature, which gave us a baseline. Applying the fundamental domain projection in this case in fact picks out one of 8 possible representatives for the Cayley table of each group, so you wouldn’t quite see exactly the same matrix. Nevertheless, it is not clear to us how one could apply some kind of checksum or look-up without using a fundamental domain projection to get 100% accuracy. For groups of order 8, the total number of Cayley Tables after permutations is 8!x5=200,000, so doing a direct look-up may be feasible; however, replace groups of size 8 with groups of size 64, say, and we would expect to see the same phenomenon as shown in our experiment, but a look-up without first performing some kind of projection becomes completely infeasible.

---

### Official Review · Reviewer_A7uP · 2022-10-18
**When does projecting the input on a fundamental domain reduce accuracy, robustness ? and how so?**

**Confidence:** 3
**Soundness:** 3
**Presentation:** 3
**Contribution:** 2
**Overall Rating:** 5

**Summary:**

The authors study an input space on which a finite group, G, acts. They want to leverage the knowledge of a discrete group acting on inputs to improve learning invariant features, invariant under the action of G (G-invariant). They propose a preprocessing step that allows building G-invariant representations of the input: the projection $\pi$ on a fundamental domain. They give algorithms to compute this projection.

They apply this method to two tasks: binary classification problem on synthetic Cayley tables, and prediction of Hodge number of a complete intersection Calabi-Yau manifold. In the second case, they improve the state of the art.

They apply their preprocessing on MNIST which is less performant than data augmentation.

**Questions:**

1) Could the author discuss when does projecting the input on a fundamental domain reduce accuracy, robustness ? and how so? how does it conflict with 'less samples implies lower accuracy'? Do the authors have more numerical results on that subject.

2) The authors state that such method could be applied to continuous groups. When doing so the dimension of the input space (the space in which the samples live) is reduced; I believe it conflicts even more with the idea that data augmentation helps for better accuracy. For example, imagine that the dataset is an open of a vector space and that the projection reduces it to a closed set with empty interior, in this case, the complexity of the network considered on the whole open set might be too big and it might overfit, leading to lower accuracies. Could the authors discuss this point?


3) How significant is predicting the second Hodge number of a CICY matrix and the result of the author?

**Limitations:**

*see Strengths And Weaknesses

**Recommended Decision:**

2: Borderline

**Relevance:**

3: Solid fit

**Strengths And Weaknesses:**


---------------------------
Originality: the contribution seems novel especially this approach (fundamental domain projection) for datasets coming from pure mathematics (CICY), although I am not an expert on machine learning for datasets coming from pure mathematics.

 This approach seems to be very natural and I wonder if previous negative results in this direction are not already known for widely used datasets such as CIFAR10, CIFAR100, TinyImageNet, ImageNet. It seems that this approach is similar to reducing the size of a dataset using heuristics and I wonder if such an approach is not known to be bad for accuracy as can be seen in the experiments the authors did on MNIST (table 3). I feel that the discussion on 'how building carefully chosen sub-dataset as training dataset can introduce bias in learning ' with respect to widely used datasets would strengthen the paper. Although I understand the scope of the paper is maybe datasets that are relevant for 'group theory, string theory'.



Quality : the submission is technically sound.

Clarity: the paper is very clear and the introduction is much appreciated especially section 1.1 on existing work on the subject.

Significance : I believe the significance of the paper is table 2 : the preprocessing proposed beats state-of-the-art on predicting the second Hodge number on CICY. I think the scope of the contribution should be made more explicit and maybe restrained : the paper results on datasets coming from mathematics are interesting (and may be significant, I can't assess) and give motivation for the approach, however, the results on image recognition are a negative result of the approach on MNIST; this means that one would not expect to get better results on higher resolution datasets such as CIFAR. For image recognition, I believe that the discussion with respect to other preprocessing methods for example data augmentation is not detailed enough; a discussion on the understanding of the authors on why their method fails on MNIST comparatively to other preprocessing methods would be a good addition to the paper.

In conclusion, it seems that the paper has two unequal results one positive on datasets coming from mathematics but negative on images, and link with existing methods in the second case is missing.  This penalizes the paper.

---------------------------

Overall comment: I believe the method is worth publishing and the negative result on MNIST is interesting but not discussed enough.

**Submission Track:**

Proceedings Paper (9 Page)

---

> ### Author Response · Authors · 2022-10-31
> **Our approach generally preserves the size of training data**
>
> Originality
> Reviewer: ‘It seems that this approach is similar to reducing the size of a dataset using heuristics’
> Reply: In general our approach is much more closely related to the idea of finding canonical or normal form for the data, rather than reducing the data by heuristics. In particular, the size of the training data stays the same after projecting, it’s just reorganised in a geometrically more sensible way. Under some rather exceptional circumstances where a lot of the training data lies in the same G orbits, our approach does have the effect of reducing the size, but this is only likely to happen in synthetic problems, such as the Cayley Table example in the paper.
>
>
> Significance
> Reviewer: “a discussion on the understanding of the authors on why their method fails on MNIST comparatively to other preprocessing methods would be a good addition to the paper.”
> Reply: We have added the following sentence to the introduction: “In the case of small group sizes, our approach makes no notable difference. That is for example the case in image recognition tasks with rotational symmetry, which is one of our examples.”
>
>
> Questions
> We reply to questions in the order they were asked
>
> (1)We have not found any cases where projecting makes the machine learning algorithm measurably worse. Heuristically, we would expect that our approach should never be worse than doing nothing except perhaps in some highly synthetic situations, but again we haven’t observed this.
>
> (2)This would need to be taken care of when working out the continuous case, this is beyond the scope of this paper.
>
> (3)We have added more details on Hodge numbers and CICYs to the beginning of 3.2.

---

### Decision · Program_Chairs · 2022-10-21

Accept (Poster)